# When Does LoRA Reuse Work? Theoretical Limits and Mechanisms for Recycling LoRAs Without Data Access

**Mei-Yen Chen** [*†2]    **Thi Thu Uyen Hoang** [*2]    **Michael Hahn** [*1]    **M. Saquib Sarfraz** [2]
[1] **Saarland University**
[2] **Mercedes-Benz Tech Innovation GmbH**

**Reviewed on OpenReview:** `https://openreview.net/forum?id=lVqUJlsnRy`

## Abstract

Reusing low-rank adapters (LoRAs) by merging or routing is a common strategy for adapting large language models to new tasks, especially when training data is unavailable but many fine-tuned LoRAs are accessible. While the availability of publicly shared LoRA weights has inspired new algorithms for composing them to solve new tasks, recent findings highlight limitations in LoRA's ability to integrate new knowledge. This work investigates when LoRA reuse can be successful for compositional factual and reasoning tasks. Through theoretical analysis in a simplified setup and experiments on a controlled synthetic two-hop reasoning task with extensions to math word problems, cross-lingual code generation, and history/geography QA, we show that data-agnostic methods, such as parameter averaging and dynamic selection, often fail to combine knowledge from logically disjoint fine-tuning datasets. This challenge is particularly pronounced when the relevant knowledge is underrepresented during pretraining. However, reuse can succeed when fine-tuning datasets share solution templates, such as reasoning patterns or reusable code, which serve as bridges among tasks. Our results suggest that LoRA reuse relies more on shallow pattern matching than on logical integration of existing knowledge. This mechanism-based perspective offers practical guidance for curating datasets and designing systems that enable LoRA reuse to overcome data-access limitations. Findings indicate that future research should focus on the mechanisms enabling effective adapter reuse rather than solely on developing new reuse algorithms.

## 1 Introduction

Many real-world applications require large language models to integrate scattered information and infer logical answers to novel questions. For instance, an AI assistant supporting human resource specialists in determining an employee's tax rate must combine information about the employee's marital status and the spouse's residency, as this affects the application of tax law. Such information often originates from scarce data sources or resides in separate systems where regulatory constraints limit direct access. Hence, compositional generalization, the ability of a model to create new combinations of known elements, is essential for the quality of such applications.

As pretraining and fine-tuning become standard practices in Large Language Model (LLM) development, reusing shared model weights from foundation models and their fine-tuned variants has emerged as a practical strategy for generalization in data-scarce scenarios. Unlike Federated Learning (McMahan et al., 2017), this so-called model merging (Raffel, 2023) approach passively operates on shared model weights without coordinated training rounds. As parameter-efficient fine-tuning (PEFT) methods gain popularity, combining fine-tuned modules, especially low-rank adapters (LoRAs) (Hu et al., 2021), has emerged as a data-free alternative to enhance model capabilities (Beck et al., 2022; Huang et al., 2024; Zhao et al., 2024b; Ostapenko

---

[*]Equal contribution
[†]Corresponding author: mei-yen.chen@mercedes-benz.com

et al., 2024; Prabhakar et al., 2024; Zhao et al., 2024a; Yadav et al., 2025). The idea is that users can exchange and merge LoRA updates at inference time, just like plug-and-play libraries in a software program. Consequently, this idea has sparked a proliferation of novel methods for reusing fine-tuned LoRA weights for new tasks (e.g. Huang et al., 2024; Ostapenko et al., 2024; Zhao et al., 2024b; Beck et al., 2022; Zhang et al., 2025).

While these approaches improve computational and economic efficiency, they are often developed and evaluated under very different experimental setups, each with its own assumptions about system architecture, data availability, usage scenarios, and computational limits. While recent work has addressed such inconsistencies in combining entire fine-tuned foundation models (Tam et al., 2024), the various design choices for merging or routing LoRA modules have only been surveyed (Yadav et al., 2025), leaving many questions unanswered. Furthermore, LLMs gain knowledge through pretraining, while supervised fine-tuning of instruction-following tasks teaches them the style or format for user interaction (Zhou et al., 2023). Consequently, fine-tuning LLMs with new knowledge often leads to hallucinations (Gekhman et al., 2024; Ghosal et al., 2024). Low-Rank Adaptations (LoRAs) are inherently limited in their expressiveness (Zeng & Lee, 2024) and can reduce chain-of-thought (CoT) reasoning abilities (Lobo et al., 2025).

In many real-world scenarios, producing high-quality answers is essential, yet data access is often limited (e.g., health care, finance, medicine). **This raises our research question: how effectively can openly shared LoRA modules be reused to generate valid answers for new tasks without direct access to the corresponding raw data?** To address this, we began with a theoretical analysis in a simplified setup that reveals limitations of LoRA reuse. This is further supported by empirical findings using LLMs (parameter sizes ranging from 1.5B to 70B) and synthetic tasks that were designed as proxies to real-world use cases: 2-hop reasoning to combine personal information and math reasoning in math word tasks. We focus on **compositional factual and reasoning tasks** in order to cleanly disentangle knowledge in the base model from the effect of composing different LoRAs: We can ensure that correct answers can only be obtained by composing knowledge from the reused LoRAs, and not by retrieving knowledge already present in the base model. Together, our methods isolate factors affecting LoRA reuse, highlight challenges, and provide practical data preparation insights for designing effective solutions in these scenarios. Our main findings are as follows:

- LoRA reuse is ineffective for new tasks regardless of model size or pretraining corpus.

- LoRA reuse can be successful without direct data access when the common solution templates (e.g., chain-of-thought reasoning patterns or reusable code snippets) existed in the fine-tuning datasets as bridges.

Overall, our findings indicate that LoRA reuse relies on shallow pattern matching rather than on logically integrating information across independently trained LoRAs. Understanding these mechanisms is essential for designing systems that can effectively reuse adapters and for preparing fine-tuning datasets that support reliable composition. Practitioners should carefully consider the target application when combining existing LoRAs, as curated training data plays a critical role in successful reuse.

In the following sections, we review related work, present theoretical analysis, and report empirical results using a controlled two-hop reasoning setup to isolate key failure mechanisms. We then validate these insights in more realistic domains, including math word problems, cross-lingual code generation, and history/geography QA. Our study thus clarifies when LoRA reuse is effective without data access and highlights the need for mechanism-driven research to guide future system design.

## 2 Related Work

LoRA modules (Hu et al., 2021) have emerged as a privacy-friendly, data-free method for sharing model capabilities. Users can exchange LoRA updates or merge them at inference time, like plug-and-play libraries in a software program (Beck et al., 2022; Huang et al., 2024; Zhao et al., 2024b; Ostapenko et al., 2024; Prabhakar et al., 2024; Zhao et al., 2024a; Yadav et al., 2025). However, many methods for reusing LoRAs

require examples from unseen tasks to estimate merging weights or routers, raising questions about how much successful generalization can be attributed to the LoRAs themselves. This highlights the need for mechanisms that ensure effective LoRA combination without data access.

Weight averaging and routing are two common strategies for reusing publicly shared LoRA weights that were trained with the original PEFT approach (Hu et al., 2021). Weight averaging has become a popular approach, motivated by several findings. First, fine-tuned models remain in the same loss basin as pretrained weights (Neyshabur et al., 2020). The difference between pretrained and fine-tuned model weights—task vectors—can further steer model behavior through arithmetic operations (Ilharco et al., 2023). Recent algorithms focus on resolving merge conflicts (TIE-Merging (Yadav et al., 2023)), randomly pruning redundant parameters (Yu et al., 2024), and estimating weights for averaging LoRAs (e.g., LoRA-hub (Huang et al., 2024)). However, the mechanisms enabling successful generalization remain unexplored. Routing is another popular alternative, inspired by the Mixture of Experts (MoE) architecture. This type of approach treats openly shared LoRA adapters as domain experts and develops algorithms to reuse them in novel domains. However, many methods require additional data to set up the MoE for expert retrieval or router training (Chronopoulou et al., 2023; Zhao et al., 2024a; Jang et al., 2023). Arrow (Ostapenko et al., 2024) is a notable exception, routing LoRAs directly based on similarity between query tokens and singular values of LoRA experts.

Beyond standard LoRA (Hu et al., 2021), recent work has explored alternative PEFT fine-tuning and system designs to derive compositional strategies. For instance, LoRI (Zhang et al., 2025) reduces interference by freezing projection matrices and applying task-specific masks. Similarly, LoRA Soups (CAT) (Prabhakar et al., 2024) freezes one low-rank matrix during fine-tuning and subsequently merges adapters by concatenating their weights, explicitly leveraging held-out data to train a routing mechanism for composite tasks. LoRA Lego (Zhao et al., 2024b) decomposes adapters into rank-level units for flexible recombination. Self-MoE (Kang et al., 2024) takes a different route, creating self-specialized experts using synthetic data and dynamic routing for task-specific activation.

**What can be recycled from a hub of LoRAs?** Despite the surge of novel algorithms, when privacy is paramount, choices remain limited, especially for zero-shot generalization without access to training data. The latent logic in the pretraining corpus or term frequency may play roles in combining LoRAs for zero-shot generalization. Scaling language models has revealed emergent abilities for zero-shot reasoning (Wei et al., 2022; Kojima et al., 2022), suggesting that LLMs learn latent logical knowledge during pretraining. Task vectors demonstrate analogical reasoning through arithmetic operations (Ilharco et al., 2023), but their effectiveness may depend on term co-occurrence frequency in pretraining data (Merullo et al., 2025). If LoRAs serve as linear approximations of fine-tuned tasks, then term-frequency effects from pretraining may constrain their ability to generalize to tasks underrepresented in the pretraining dataset. Another possibility is that observed generalization performance via merging or routing LoRAs reflects superficial pattern matching rather than genuine compositionality. Empirical studies indicate that LLMs rely on token-level cues, with small lexical changes affecting reasoning performance (Mirzadeh et al., 2024; Li et al., 2024a). LLMs struggle with latent multi-hop reasoning, relying on explicit prompting to bridge logical gaps (Press et al., 2023; Balesni et al., 2025). Synthetic reasoning tasks therefore play a key role in assessing compositional generalization and indicate how effectively LoRA combination transfers to entirely novel tasks.

**When does reusing fine-tuned adapters work without training data?** This challenge is particularly relevant in domains such as healthcare, finance, and medicine, where data sharing is restricted but accuracy is essential. To ensure practical relevance, our study focuses on standard LoRA (Hu et al., 2021), the predominant PEFT method and the default in numerous open-source frameworks and cloud platforms. Our investigation begins with a theoretical analysis of two-hop reasoning that reveals limits of reusing LoRA weights. We then present experiments using synthetic data as proxies for real-world scenarios under data constraints. The experiments were conducted on LLMs with diverse pretraining histories and parameter scales (1.5B–70B) to identify conditions for successful LoRA reuse. Finally, we extend the study to math word problems to examine how data augmentation strategies (e.g., chain-of-thought templates, reusable code snippets via multi-agent generation) and layer-targeted fine-tuning can further facilitate success. The next section details our methodology and experimental design.

# 3 Theoretical Analysis

Here, we argue theoretically that low-rank adaptation, while it can store new facts in transformers, is unlikely to lead to compositional behavior when combining different LoRAs. We study this by considering the problem of composing knowledge from two LoRAs, where each contains factual knowledge, and their combination is expected to perform two-hop reasoning (e.g. Yang et al., 2024c; Balesni et al., 2025) that requires both pieces of knowledge. In general, direct theoretical understanding of multi-layer softmax transformers is very difficult; but many theoretical insights have been obtained by studying simplified setups, especially one-layer models and the limit of very wide models (Hron et al., 2020). We use this approach to perform a simple analysis of low-rank adaptation for factual knowledge. Our theoretical analysis is grounded in the study of Nichani et al. (2025), who develop a theoretical model of storage and recall of relational facts in transformers, in the simplified setup of a one-layer transformer. One-layer transformers are a popular model in the theoretical literature, being more tractable than multi-layer models while offering useful insight (e.g. Peng et al., 2024; Li et al., 2024b; Sanford et al., 2023; Tian et al., 2023). Nichani et al. (2025, Section 4) show that either an MLP or an attention head can perform factual recall. We focus our analysis on factual recall via MLPs, both because they are more expressive than attention-only retrieval, and because interpretability work suggests MLPs are important in factual recall (Geva et al., 2021; Meng et al., 2022; 2023). We discuss the detailed comparison of our setup to that of Nichani et al. (2025) in Appendix A.1.2.

Adapting an MLP performing factual recall with LoRA on a one-hop prompt can change individual facts – such as setting $r_1(x_1)$ to $x_2$. However, importantly, combining LoRAs adapting two relations on such an MLP will not result in compositional behavior, as we show below. As a consequence, a simplified one-layer transformer will not be able to show compositional behavior when combining LoRAs. While the theoretical model applies to a simplified one-layer setting, our empirical results (Section 4.1) confirm that this conclusion also holds for real-world LLMs, even though they have many layers.

## 3.1 Setup

**Entities, Facts, and Prompts** We consider a set $\mathcal{X}$ of entities, and a set $\mathcal{R}$ of binary relations $r \subset \mathcal{X} \times \mathcal{X}$ (e.g., X is married to Y; Y lives in Z, etc). We assume that each $r$ is a partial function, i.e., for each $x$, there is at most one $y$ satisfying $(x, y) \in r$; we write $y = r(x)$. Whereas Nichani et al. (2025) relied on the assumption that each relation maps to a disjoint output space, we avoid this assumption. We assume that the model operates on the following types of prompts

1. One-Hop: `X REL` (where `X` represents an entity $x \in \mathcal{X}$ and `REL` represents a relation $r \in \mathcal{R}$, with expected completion: `Y`, where $y = r(x)$ (e.g., "the spouse of X is Y").

2. Two-Hop: `X REL1 REL2` (where `REL1`, `REL2` represent relations $r_1, r_2 \in \mathcal{R}$), with expected completion: `Y`, where $y = r_2(r_1(x))$ (e.g. "the place of birth of the spouse of X is Y").

**Simple Transformer Model** We consider a vocabulary consisting of relations $r_1, r_2, \ldots$ and entities $x_1, x_2, \ldots$; with token embeddings $e_{r_i}, e_{x_i} \in \mathbb{R}^d$. We will write $E \in \mathbb{R}^{|\mathcal{X} \cup \mathcal{R}| \times d}$ for the matrix holding all token embeddings. We assume a single softmax attention head with $K, Q, V \in \mathbb{R}^{d \times d}$ matrices, and a ReLU MLP with hidden dimension $m$ given by matrices $U \in \mathbb{R}^{m \times d}$; $W \in \mathbb{R}^{|\mathcal{X}| \times m}$, mapping a vector $x$ to $W \cdot \phi_{ReLU}(Ux)$. We assume strict positional masking without attention to the current position (e.g. Yang et al., 2024b; Li & Cotterell, 2025), and a residual connection. We do not require positional encodings. We assume that the next-token prediction is provided by $W$ as a one-hot encoding of the target entity, $i_x$, omitting softmax for simplicity. Our aim is to showcase limitations in composition, not in storage of knowledge itself; hence, we allow the model a width $d$ substantially larger than $|\mathcal{X}|, |\mathcal{R}|$. In order to give the MLP as much capacity as needed, we allow $m$ to be arbitrarily large.

Nichani et al. (2025) took $E$ to be randomly initialized and not trained. We follow this assumption, and additionally take $U, V, K, Q$ to remain untrained. Overall, we assume that $U, V, K, Q, E$ matrices are randomly initialized, all with entries from $\mathcal{N}(0, \frac{1}{d})$. We focus consideration of training to $W$. This represents a random features setup (e.g. Rahimi & Recht, 2008; Ghosal et al., 2022; Dirksen et al., 2022). In this setup, softmax

attention is close to uniform; we will take it to be exactly uniform for simplicity. We further remove the softmax over the vocabulary in the output. We emphasize that training only $W$ is a substantial simplification, which makes theoretical analysis tractable. The training analysis in Nichani et al. (2025) makes a different set of simplifications (linear MLP and linear attention); we opt to keep a nonlinear MLP due to its greater expressiveness at representing relational knowledge, at the cost of assuming a random features setup. In both cases, simplifications amount to keeping only one nonlinearity (ReLU in our case; softmax over the vocabulary in Nichani et al. (2025)) and focus trainable parameters on one side of the nonlinearity (above it in our case, below it in Nichani et al. (2025)). We provide further discussion in Appendix A.1.2.

We will examine the situation where the base model already performs correctly for the given set of relations $\mathcal{R}$, and $W$ is then adapted to reflect edits to such facts.

We focus LoRA on $W$, in agreement with our experimental finding in Section 4.1 that applying to MLPs can be sufficient to get most of the gains. We consider updates $\Delta W = AB^T$ with $A \in \mathbb{R}^{|\mathcal{X}| \times s}, B \in \mathbb{R}^{m \times s}$ where $s$ is small, subject to an L2 penalty $\|A\|_F^2 + \|B\|_F^2$. We particularly consider one-rank updates, $\Delta W = pq^T$ where $p \in \mathbb{R}^{|\mathcal{X}|}, q \in \mathbb{R}^m$.

We note that this setup simplifies many aspects of transformers: there is only one layer and one head, and training focuses on the (linearized) output. We also remove the softmax over the vocabulary in the output. Our setup is designed to be *simplest possible* setup in which a nontrivial statement about LoRA's ability to learn and combine abilities can be made.

## 3.2   Results

The correct responses to all 1-hop and 2-hop relations can jointly be coded into $W$ when $d$ and $m$ are sufficiently large, due to the separation ability of the random features model (Ghosal et al., 2022). This analysis is in line with mechanistic studies of factual recall suggesting MLPs act as key-value storage (Geva et al., 2021).

Changing a fact $y = r(x)$ requires changing the output of the MLP on the subspace spanned by the entity and relation. When the update affects only a single fact, L2 regularization ensures that it has a simple and interpretable closed form:

**Proposition 1.** *A rank-one update to $W$ changing the output on a prompt* X REL *from $r(x)$ to $\tilde{r}(x)$ must have the form:*

$$\Delta W_{r \mapsto \tilde{r}} = \frac{1}{\|\phi_{ReLU}(UVe_X + Ue_{REL})\|_2^2}(i_{\tilde{r}(x)} - i_{r(x)})\phi_{ReLU}(U \cdot Ve_X + U \cdot e_{REL})^T \tag{1}$$

Here, $Ve_X$ reflects the output of attention to the preceding position, and $e_{REL}$ reflects the residual connection. This is similar to the RoME update (Meng et al., 2022). Intuitively, based on the idea that MLPs act as key-value storage, the LoRA update $\Delta W = AB^T$ specifically addresses the encoding of the prompt X REL in the $B$ matrix, and the changed output in the $A$ matrix. The proof is in Appendix A.1.

Now consider a two-hop prompt X REL1 REL2, intended to denote the composition of the two relations. Given sufficient width, any set of such two-hop facts can be encoded in $W$. However, as we next show, adding two LoRAs modifying two relations $(\Delta W_{r_1 \mapsto \tilde{r}_1}, \Delta W_{r_2 \mapsto \tilde{r}_2})$ will not unlock compositional behavior on the new facts:

**Theorem 2.** *Assume LoRAs $\Delta W_{r_1 \mapsto \tilde{r}_1}, \Delta W_{r_2 \mapsto \tilde{r}_2}$ are created to adapt two single facts for $r_1$, $r_2$. Summing these adapters will not result in correct results for composition of the two relations $r_1, r_2$.*

The formal proof is in Appendix A.1. The reasoning is as follows. As shown in Proposition 1, the two LoRAs specifically modify the MLP output on the subspaces inhabited by the activations computed on the two one-hop prompts. When the model encounters a two-hop prompt, the activations will partly overlap with the subspaces for both one-hop prompts, and the adapters will lead the model to output not the composition $r_2(r_1(x))$, but a linear mixture of two relevant entities. A natural question is whether some of the routing or weighting methods proposed in the literature resolve this; it turns out that the argument extends to those: For instance, weighted averaging of the two adapters (e.g. Prabhakar et al., 2024; Ostapenko et al., 2024) will

still fail to perform compositionally when several facts are updated. Formally (see Appendix A.1.1 for the proof),

**Theorem 3.** *The same result holds for CAT (Prabhakar et al., 2024), linear merging of LoRAs (Yadav et al., 2023; Yu et al., 2024; Huang et al., 2024), Arrow routing (Ostapenko et al., 2024).*

Yet another approach might be to combine a larger library of LoRAs where some have been trained on 2-hop examples from other task pairs. One might hope that this would prime the model towards compositional behavior; however, the reasoning above still applies, and suggests that reusing LoRAs would still fail to behave compositionally (Appendix A.1.1).

One limitation of our theoretical analysis is that (in line with Nichani et al. (2025)) it applies to a single-layer transformer; our experiments test applicability of the conclusions to LLMs across scales.

## 4 Experiments

The following experiments aim to apply the above theoretical analysis in full-scale LLMs to determine under what conditions combining LoRAs enables LLMs to perform new tasks that requires logical integration of information segmented among fine-tuning datasets. We focus on two data-agnostic routing methods, Uniform averaging and Arrow routing (Ostapenko et al., 2024), which operate directly on LoRA weights (see Appendix A.2.2). We designed two synthetic tasks, two-hop reasoning and math word problems, as proxies for real-world cases that require combining personal information or mathematical reasoning under data-scarce conditions. This helps identify key preconditions for effective LoRA routing: entity familiarity, domain-specific pretraining, and common solution templates that link segmented information across fine-tuned adapters. We further examine how these effects hold in different routing strategies and scale across base model sizes ranging from 1.5B to 70B parameters. To rigorously validate the generalizability of these findings, we extend our analysis to two realistic domains: cross-lingual code generation and history/geography question answering (Appendices A.5 and A.6).

### 4.1 Two-Hop generalization

We investigate whether combining two LoRAs enables compositional reasoning. Building on our theoretical analysis and inspired by Balesni et al. (2025), we design a two-hop reasoning task requiring composition across linguistic variations while controlling for base model knowledge. The dataset uses a fixed structure:

1. **First Hop** ($A \rightarrow B$), identifying the spouse of a given entity $A$,

2. **Second Hop** ($B \rightarrow C$), identifying the residence of $B$,

with the goal of inferring $A \rightarrow C$ (identifying the residence of the spouse of a given entity). This setup closely follows our Theorem 2, which suggests that LoRAs trained on one of the two hops each would, if combined, not unlock the indirect relationship.

We conduct three datasets varying the nature of entity names and locations while ensuring that the relational facts remain synthetic: $F$ (fake names, fake locations), where both entities and locations are synthetic (e.g., $(Zint, Frosk, Narik)$); $H$ (fake names, real locations), where names are synthetic but locations are real (e.g., $(Zint, Frosk, London)$); and $R$ (real names, real locations), where both names and locations are real, but relationships are deliberately shuffled to remain false (e.g., $(Barack\ Obama, Camila\ Alves, London)$). We refer to the first-hop ($A \rightarrow B$), second-hop ($B \rightarrow C$), and the two-hop ($A \rightarrow C$) subsets of each dataset as $F_1, F_2, F_{12}, H_1, H_2, H_{12}$, and $R_1, R_2, R_{12}$, respectively (see Table 5 and Appendix A.2.1 for examples and details).

Based on ablation studies (Table 8 and 9 in Appendix A.2.2), we focused on fine-tuning only the MLP layers of the following base models: Qwen2.5-3B-Instruct, Qwen2.5-7B-Instruct, and Qwen2.5-14B-Instruct (Qwen-team, 2025), as well as DeepSeek-R1-Distill-Qwen-7B, DeepSeek-R1-Distill-Qwen-14B, and DeepSeek-R1-Distill-LLaMA-70B (DeepSeek-AI-team, 2025).

### 4.1.1 Impact of base model and familiarity

**Experiment setup** For each dataset ($F$, $H$, $R$), we train four LoRA adapters (experts) LoRA 1, LoRA 2, LoRA 3 (Oracle Expert), and LoRA 4 (Mixed Two-Hop Expert) on relation $A \rightarrow B$, $B \rightarrow C$, $A \rightarrow C$, and mixed data ($A \rightarrow B$, $B \rightarrow C$), respectively. From these experts, we construct two libraries: **2-combination library** which includes LoRA 1 and LoRA 2; and **3-combination library** which includes LoRA 1, LoRA 2, and LoRA 3. We evaluate the model's ability to generalize and infer $A \rightarrow C$ relationships. We use Chain-of-Thought (CoT) prompting during testing for the 3-combination library and the 2-combination library.

**Results and analysis** As shown in Figure 1a, performance on the 3-combination library for the $H$ dataset improves with model size, while the 2-combination library remains consistently poor. Figure 1b shows that $R$ outperforms $H$, both far exceeding $F$. Notably, with only $A \rightarrow B$ and $B \rightarrow C$ adapters (2-combination library), accuracy stays below 10%, supporting Theorem 2 that composing knowledge across separate LoRAs is inherently challenging. Even when $A \rightarrow C$ is covered (3-combination library), routing does not always succeed, in particular in smaller models and the presence of of unfamiliar entities, such as fake names or cities in the $F$ dataset. These trends hold across datasets and model families (see Table 7 in Appendix A.2.2).

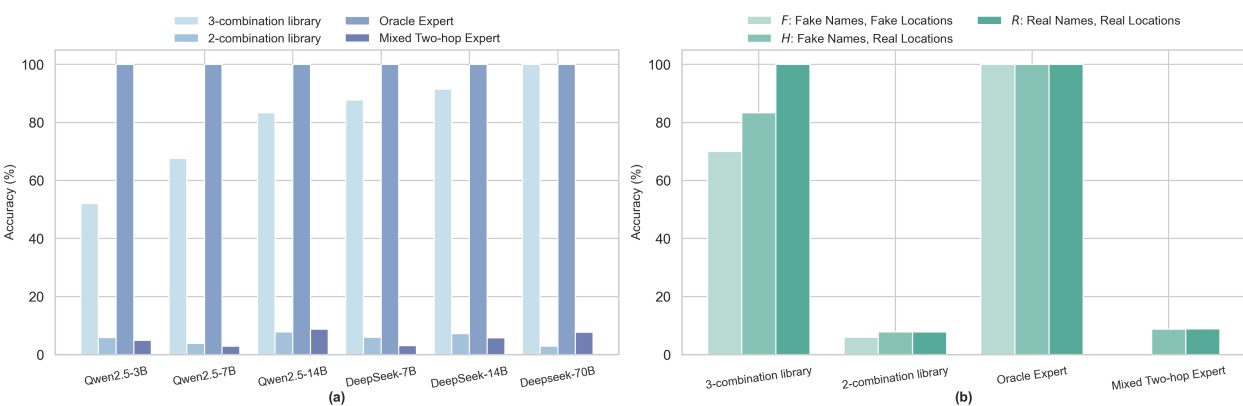

Figure 1: LoRA reuse is ineffective across model sizes and pretraining corpora, as shown by performance comparisons between a library of multiple LoRA adapters and individually fine-tuned LoRA experts on two-hop datasets. (a) Comparison of library-level and expert-level performance on the test set of $H$ across different base models. (b) Impact of entity familiarity on the performance of LoRA combination methods, using Qwen2.5-14B-Instruct as the base model, evaluated on three test sets: $F$, $H$, and $R$. Across setups and model sizes, performance on the 2-combination library is poor; including an expert trained on the target task ($A \rightarrow C$ relationship) is necessary.

### 4.1.2 Composition requires close match between testing and training prompts

So far, we found that synthesizing two-hop reasoning from two LoRAs is difficult, in agreement with our theoretical predictions. What strategies could enable composition? While a substantial amount of work has found chain-of-thoughts to support reasoning (including in the 2-hop setup when employing full-parameter finetuning, (Balesni et al., 2025)), we found that composition from the two $A \rightarrow B$ and $B \rightarrow C$ experts performs poorly even with chain-of-thought prompting. Our theoretical analysis suggests that, as LoRA adapters rely on targeting specific low-dimensional subspaces, compositional behavior can only be unlocked when the target prompts show a close formal match to prompts on which the LoRAs were trained. CoT prompting might thus be insufficient as the form of the targets mismatches the one-hop training examples of the LoRAs. However, mixing in CoT examples into the LoRA training data might be sufficient. In this section, we test if it is possible to enable composition by including CoT templates in the training data, and how close the match in reasoning patterns needs to be between finetuning and testing datasets. We specifically define the following *bridge* technique, and design a series of experiments to determine how closely the target task must be represented in the LoRA fine-tuning datasets to enable compositional behavior (Figure **??** left

panel). The idea is that the finetuning dataset additionally includes examples of the targeted reasoning pattern. Via ablations, we test which aspects of the targeted reasoning pattern needs to be present.

**Experimental setup**   We design two *bridge* variants over the $F$ and $R$ datasets (Figure 2, left panel). The **Fake Bridge** ($B_F$) is constructed by concatenating the $F_1$ ($A \to B$), $F_2$ ($B \to C$), and $F_{12}$ CoT (CoT-formatted $A \to C$) subsets. The **Real Bridge** ($B_R$) follows the same structure but uses the corresponding subsets ($R_1$, $R_2$, $R_{12}$ CoT) from the $R$ dataset, which contains real names and locations (see Table 6 and Figure 3 in the Appendix A.2.1 for examples and details). We fine-tune adapters on the union of direct-hop examples and a bridge dataset. In the first configuration, **Setup 1**, models are trained by mixing fake data subsets ($F_i$) with the Real Bridge set ($B_R$): LoRA 1 is trained on $F_1 + B_R$, and LoRA 2 on $F_2 + B_R$, with evaluation on the held-out subset $F_{12}$. In the second configuration, **Setup 2**, models are trained by mixing real data subsets ($R_i$) with the Fake Bridge set ($B_F$): LoRA 1 is trained on $R_1 + B_F$, and LoRA 2 on $R_2 + B_F$, with evaluation on the held-out subset $R_{12}$.

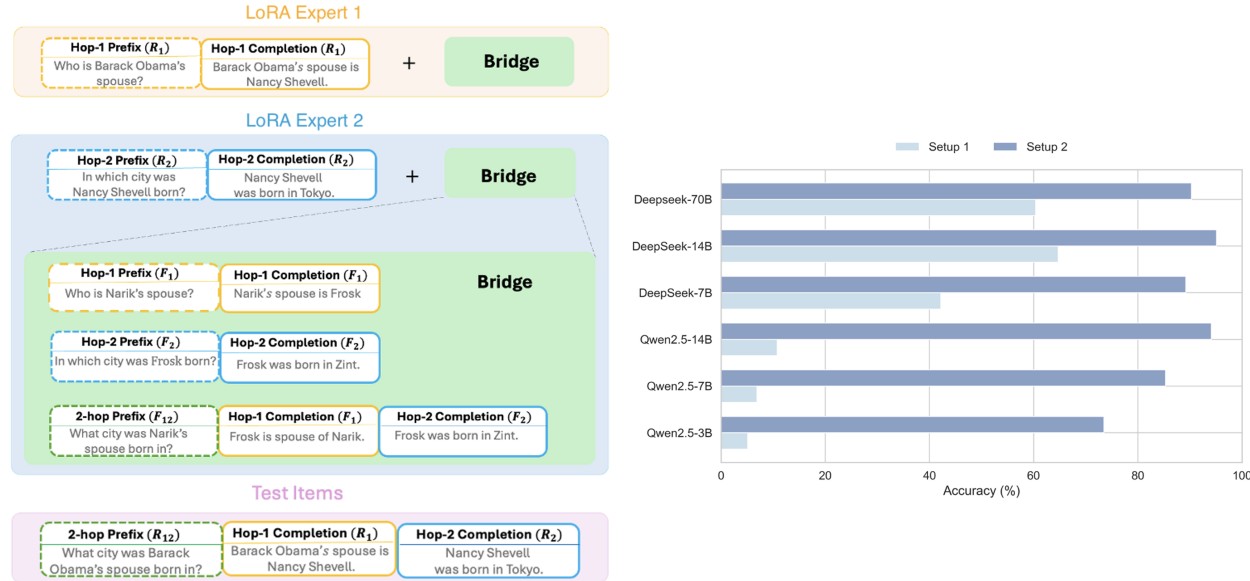

Figure 2: (left) In the bridge setup, both LoRA experts are trained not only on one of the two hops, but also on examples (disjoint from those needed in testing) of both hops, and chain-of-thought two-hop reasoning. (right) Performance comparison of two setups across different base models: Setup 1 (**Real Bridge**, $B_R$) adds a bridge using real names and locations to a dataset using fake names and fake locations ($F$), while Setup 2 (**Fake Bridge**, $B_F$) reverses these. Each setup uses LoRA 1 and LoRA 2 as the experts in the library. The bridge setup is much more successful in Setup 2.

**Results and analysis**   Figure 2 demonstrates that explicitly incorporating the target two-hop reasoning pattern into the LoRA fine-tuning data is crucial for achieving reliable compositional generalization. In **Setups 1** and **2**, where each adapter is trained on a synthetic direct-hop subset combined with the bridge dataset, *Arrow* performance improves significantly compared to earlier experiments. Additionally, the bridge setup is much more successful in **Setup 2**, highlighting the importance of entity familiarity for effective generalization.

We conducted a set of ablations to analyze which aspects are important for the success of this strategy. Incorporating structured reasoning into LoRA finetuning yields only marginal gains unless the finetuning data closely mirror the target two-hop task. First, as shown in Table 1, the bridge improves two-hop accuracy only when CoT-formatted $A \to C$ instances are included during adapter training. Omitting CoT formatting results in worse performance (**Setups 2** vs. **3**). Second, simply including the bridge in only one of the two LoRA adapters (**Setups 4, 5**), or providing only $A \to C$ prompts without the individual one-hop tasks (**Setups 6, 7**), results in significantly weaker compositional performance compared to setups where both $F_1$

($A \rightarrow B$) and $F_2$ ($B \rightarrow C$) are included alongside CoT bridging. This highlights the importance of exposing the model to each subtask. Third, we found that relaxing the bridge to use disjoint task pairs still produces nontrivial gains (**Setup 8**, which uses completely different relations as the bridge dataset: $F_4$: *study_in* and $F_5$: *child_of*, see Table 5 in the Appendix A.2.1 for examples), suggesting that exact task-pair matching is less critical so long as the finetuning set contains examples reflecting the overall reasoning pattern. Altogether, these results confirm the importance of CoT exemplars and the individual tasks to unlocking generalization of Arrow routing, even if the examplars are semantically different from the target task.

Aside from combinations of two or three LoRAs, we further tested what happens when increasing the number of tasks present in the collection of LoRAs, including both various one-hop tasks, and also various two-hop tasks. Even in this case, we found that composition was very difficult, again in agreement with our theoretical predictions (Analysis in Appendix A.3).

Table 1: Ablations for the bridge training setups. We compare the 2-combination libraries trained on just the two hops (0), the full bridge (2), with various strategies interpolating between these, such as providing a bridge in only one expert (4 and 5), or omitting the CoT template from the bridge (3). The full bridge attains highest performance, and versions not including a bridge CoT in both of the two experts show poor performance (0, 3, 4, 5).

| Setup | LoRA 1 | LoRA 2 | Qwen2.5-3B-Instruct | Qwen2.5-7B-Instruct | Qwen2.5-14B-Instruct | DeepSeek-R1-Distill-Qwen-7B | DeepSeek-R1-Distill-Qwen-14B | DeepSeek-R1-Distill-Llama-70B |
|---|---|---|---|---|---|---|---|---|
| 0 | $R_1$ | $R_2$ | 1% | 3.9% | 7.8% | 8.8% | 3.9% | 5.9% |
| 2 | $R_1$ $F_1 + F_2 + F_{12}$ CoT ($B_F$) | $R_2$ $F_1 + F_2 + F_{12}$ CoT ($B_F$) | **73.5%** | **85.3%** | **94.1%** | **89.2%** | **95.1%** | **90.3%** |
| 3 | $R_1$ $F_1 + F_2 + F_{12}$ | $R_2$ $F_1 + F_2 + F_{12}$ | 1% | 2.9% | 3.9% | 3.9% | 11.8% | 3.9% |
| 4 | $R_1$ $F_1 + F_2 + F_{12}$ CoT ($B_F$) | $R_2$ | 9.8% | 8.8% | 11.8% | 8.8% | 12.7% | 10.2% |
| 5 | $R_1$ | $R_2$ $F_1 + F_2 + F_{12}$ CoT ($B_F$) | 10.8% | 35.3% | 27.5% | 25.9% | 22.5% | 26.1% |
| 6 | $R_1$ $F_{12}$ CoT | $R_2$ $F_{12}$ CoT | 40.2% | 61.8% | 65.7% | 75.5% | 72.5% | 87.3% |
| 7 | $R_1$ $F_1 + F_{12}$ CoT | $R_2$ $F_2 + F_{12}$ CoT | 27.5% | 75.5% | 79.4% | 74.5% | 76.9% | 89.9% |
| 8 | $R_1$ $F_1 + F_4 + F_{14}$ CoT | $R_2$ $F_5 + F_2 + F_{52}$ CoT | 43.1% | 82.4% | 92.2% | 81.6% | 83.5% | 87.9% |

We also investigate on how the 2-hop reasoning gap can be closed without reasoning templates as bridge but alternative merging methods or extra training when data access would be allowed. To achieve this, we utilize LoRA 1 and LoRA 2 in Setup 0 as shown in the Table 1 and apply uniform averaging (Huang et al., 2024; Ostapenko et al., 2024), TIES-Merging (Yadav et al., 2023) and LoRA Soups (CAT) (Prabhakar et al., 2024) on them as comparison baselines because they represent widely discussed strategies for merging openly shared fine-tuned LoRA weights. TIES-Merging applies a structured procedure—trimming low-impact parameters, resolving sign conflicts, and selectively merging—to reduce interference when combining multiple model weights. CAT altered the original LoRA (Hu et al., 2021) by only allowed one low-ranked matrices to be fine-tuned, concatenation multiple LoRA model weights, and requires access to held-out data for routing training to composite multiple tasks. Table 2 shows that when held-out data is available for routing training, CAT achieves higher accuracy than methods operating without data access. However, its performance remains below setups that route openly shared LoRAs originally fine-tuned on well-designed datasets (Table 1, setup 2,6,8). We additionally evaluated LoRI Zhang et al. (2025) on our two-hop setting. LoRI modifies the LoRA parameterization to reduce interference when combining multiple adapters, and has been shown to improve multi-task adapter merging. As shown in Table 2, all LoRI variants (LoRI-D/S with concatenated or linear merging) achieve accuracies close to chance and clearly underperform CAT. In LoRI's linear merging, a single adapter is obtained by taking a fixed linear combination of the task adapters without using any two-hop data; our Uniform baseline is simply the special case where all tasks receive the same fixed weight. In LoRI's concatenated merging, the task-specific adapters are instead stacked into a larger block-structured adapter so that each task mainly occupies its own subspace, but this merge is still fixed and data-free.

Overall, these results support our conclusion that (i) direct composition of knowledge from different LoRAs is very difficult, even when using more sophisticated adapter architectures such as LoRI.(ii) the LoRA finetuning datasets must contain examples closely matching the target reasoning behavior.

Table 2: Performance of LoRA/LoRI-based routing and merging methods using a 2-combination adapter library on Qwen2.5-7B evaluated on the R dataset.

| Adaptation | Routing / Merging | Accuracy |
|---|---|---|
| LoRA | Uniform | 3% |
| LoRA | Arrow | 3.9% |
| LoRA | TIES | 3.8% |
| LoRA | CAT | **21%** |
| LoRI-D | Concat | 13% |
| LoRI-D | Linear | 1% |
| LoRI-S | Concat | 7% |
| LoRI-S | Linear | 3% |

### 4.2 Generalization from Easy to Hard Math Word Problems

To evaluate whether our findings hold in more realistic settings, we use the GSM-Symbolic benchmark (Mirzadeh et al., 2024), which enables controlled assessment of reasoning robustness in math across well-defined difficulty levels. Each LoRA expert was fine-tuned on GSM-Symbolic (original) and GSM-P1 (with one added clause) individually, before being combined for evaluation on GSM-P2 (which adds another clause). We compare general-purpose and math-specialized models to assess the impact of pretraining. Similar to exposing LoRAs to solutions closely resembling the target task, we also tested whether fine-tuning with reusable Markdown and Python code (Suzgun et al., 2025) would improve generalization on GSM-P2. Detailed experimental design, fine-tuning, and evaluation procedures can be found in Appendix Section A.4.1.

**Limitations of LoRA reuse.** The effectiveness of LoRA routing is highly dependent on the base model's domain-specific pretraining history. To start, we replicated the findings of Mirzadeh et al. (2024), which show that large language models (LLMs) lack robustness in mathematical reasoning (see Appendix Section A.4.2, Table 13). Routing methods such as Uniform and Arrow provided modest improvements for the general-purpose Qwen2.5-1.5B-Instruction model, but often degraded performance in math-specialized models like Qwen2.5-Math-Instruction, regardless of model size (Table 3). Among these, Uniform consistently outperformed Arrow. Echoing prior work showing that 8-shot GSM8K in-context examples do not improve performance on GSM-P2 (Mirzadeh et al., 2024), we further observed that combining these examples with LoRA routing actually worsened results. For example, in the Qwen2.5-Math-7B-Instruction model, Arrow routing with in-context examples reduced GSM-P2 accuracy from 0.27 to 0.06 (see Appendix Section A.4.2, Table 14 for details).

The performance drop observed after LoRA routing may stem from a mismatch between the fine-tuning data and the base model's capabilities. Qwen2.5-Math-Instruction is designed to solve problems using Markdown and Python code, while the GSM-Symbolic benchmarks provide only natural language Chain-of-Thought (CoT) solutions. As a result, routing LoRAs fine-tuned on this dataset may suppress the model's tool-integrated reasoning abilities and lead to an increase in calculation errors. Our error analysis follows the definitions and procedures outlined by Zhong et al. (2025). See Appendix Section A.4.2 and Table 15 for details.

Table 3: Accuracy comparison on zero-shot GSM-P2 after routing LoRA experts individually fine-tuned on GSM-Symbolic and GSM-P1.

| LoRA Routing Methods | Qwen2.5-1.5B-Inst | Qwen2.5-Math-1.5B-Inst | Qwen2.5-Math-7B-Inst |
|---|---|---|---|
| Base model only | 5% | 47% | 68% |
| Uniform | 10% | 24% | 34% |
| Arrow | 9% | 19% | 27% |

**How can programming language bridge the generalization gap?** This experiment extends our bridge results in two-hop reasoning (Table 1) to math word problems. The experimental design is motivated

by recent findings, Dynamic Cheatsheet, which demonstrate that encouraging language models to retain and apply reusable intermediate solutions during inference significantly improves their performance on math problems (Suzgun et al., 2025). We extend this idea using the GSM-Symbolic benchmark (Mirzadeh et al., 2024), where generalization from easier to harder problem variants requires understanding the full computational graph (Appendix Figure 5). In the previous setting, each LoRA is fine-tuned on partial solutions corresponding to subsets of reasoning steps (e.g., the black or orange subgraphs in Appendix Figure 5). However, routing these LoRAs alone does not suffice to solve the more complex P2 variant, which involves the complete computational graph (blue subgraph in Appendix Figure 5). We hypothesize that reusable Markdown and Python solutions can bridge partial representations and enhance compositional generalization through LoRA routing. To test this, we implemented two agent-based actor-critic workflows (Wu et al., 2024) to automatically generate reusable code snippets as fine-tuning data (See Appendix A.4.1 for implementation details). Table 4 demonstrate modest improvements in solving the complex P2 problems via routing LoRAs fine-tuned with these reusable code snippets. Such improvement is clearer in smaller model (Qwen2.5-Math-1.5B-Instruction) when fine-tuning targeted the MLP layers. This result aligns with our theoretical analysis and findings from the two-hop reasoning task. It highlights the importance of the mechanistic understanding of how to reuse LoRA adapters effectively for guiding data generation, and shows that such reuse works best when target tasks are clearly defined in advance.

Table 4: Enhancing easy-to-hard generalization by leveraging Tool-Integrated Reasoning (TIR) prompt and fine-tuning with reusable code.

| Base model | Fine-tuned Modules | Routing Methods | LoRA(GSM-Symb) and LoRA(GSM-Symb-P1) | Base Model Only |
|---|---|---|---|---|
| Qwen2.5-Math-1.5B-Inst | attention | Uniform | 14% | 18% |
| | | Arrow | 14% | |
| | MLP | Uniform | **20%** | |
| | | Arrow | 13% | |
| Qwen2.5-Math-7B-Inst | attention | Uniform | **40%** | 47% |
| | | Arrow | 24% | |
| | MLP | Uniform | 39% | |
| | | Arrow | 37% | |

## 5 Discussion

Our findings indicate that combining LoRAs is ineffective for new tasks unless common solutions to those tasks are already represented in the fine-tuning datasets. Although alternative PEFT methods may offer better compositional results, our study focuses on merging adapters without access to further training data. Allowing training would enable reconfiguration of adapters and task setups, making the reuse problem trivial. While a comprehensive empirical comparison is beyond our scope, we discuss these methods using our theoretical and empirical insights to guide future directions. For instance, LoRI (Zhang et al., 2025) addresses cross-task inference by combining random projections with task-specific masks, potentially enabling better adapter routing. However, positive results for compositional reasoning have not been reported, and our theoretical analysis suggests that it remains challenging. Similarly, LoRA Lego (Zhao et al., 2024b) formalizes low-rank updates as independent units and clusters these into new adapters to reduce interference, though it has not been shown to enable compositional reasoning. Self-MoE (Kang et al., 2024) constructs experts based on self-generated specialization training data and a trained router, but it remains unclear to what extent this method can enable compositional combinations of different abilities. FLiX (Sun et al., 2024) learns different low-rank updates for various task or dataset features, and CS-ReFT (Sun et al., 2024) learns orthonormal task-specific adapters. Despite these innovations, none have demonstrated effective compositional combination of skills, consistent with our theoretical analysis indicating inherent limitations.

Recent work (Prabhakar et al., 2024) has proposed LoRA concatenation as an effective method for composing skills to solve challenging math word problems, such as those in GSM-Hard (Gao et al., 2023). We acknowledge the significance of these findings, particularly their demonstration that decomposing skills into reusable LoRAs and estimating combination weights can enhance performance, provided that additional task-specific data and knowledge are available. However, our work takes a different perspective. Unlike GSM-Hard (Gao et al., 2023), which primarily modifies numerical ranges while preserving the question format of the

original GSM8K problems, GSM-Symbolic-P2 (Mirzadeh et al., 2024) presents more realistic and difficult compositional generalization challenges. It alters both the question format and the structural complexity of math problems into entirely unseen forms. Our theoretical analysis shows limitations (Appendix A.1.1), supported by empirical results showing that training provides little benefit in a 2-hop reasoning setting (Appendix Table 2). This suggests that the benefits of such approaches may not extend to more challenging generalization tasks like GSM-Symbolic. While skill composition remains important, our results highlight a key limitation of LoRA routing approaches: their effectiveness often depends on prior knowledge or training data of the downstream tasks, which may not be viable in practice.

A critical impediment to LoRA reuse identified in our study is negative transfer, wherein merging adapters can suppress the base model's latent capabilities or degrade a dominant skill. For example, as shown in Table 15 (Appendix A.4), the base model uses code-based responses in roughly 12% of cases, but the merged models (Uniform/Arrow) reduce this to 0%, reverting entirely to natural language. This suggests that merging may suppress specialized "tool use" capabilities, biasing the model back toward the dominant modality (text) of the individual LoRAs. Similarly, in cross-lingual code generation, merging an English-Coding adapter ($L_1$) with an English–Spanish Translation adapter ($L_2$) frequently results in lower performance than using $L_2$ alone (e.g., 28.8% vs. 36.0% on Phi-2; Appendix A.5). In this setting, the unaligned $L_1$ adapter interferes with the translation capabilities of $L_2$ when processing Spanish inputs, rather than composing to form a Spanish-coding capability. These observations highlight that LoRA merging can introduce negative transfer effects, particularly when component adapters are unaligned or target different modalities, limiting the effective compositional reuse of independently trained adapters.

The bridge data used in our experiments serves a diagnostic purpose: delineating the boundaries of naive composition rather than supplying auxiliary supervision. By demonstrating that reuse succeeds only when training data structurally mirrors the target task, these experiments confirm that pure data-free reuse is constrained by prompt alignment. This insight defines the representational overlap required for modularity, suggesting that future efforts should focus on curating bridge-rich datasets to enable interoperability.

The success of bridge datasets implies that the LoRA reuse strategies tested in this work operate primarily through template exposure—priming the model to output in a specific format or style—rather than by composing disjoint knowledge bases. Although our evaluation tasks focused primarily on factual knowledge and logical reasoning, we hypothesize that other applications such as adapting a model's style (e.g., adopting a specific persona) or format (e.g., outputting JSON) via reuse may be viable without bridge datasets, as these rely on surface-level templates. However, deep knowledge generalization (e.g., combining disjoint "Legal" and "Medical" LoRAs) likely fails without bridging data. Future research should rigorously test this boundary.

## 6 Conclusions

Our experiments examined when combining LoRAs enables LLMs to perform new tasks by integrating capabilities from different adapters. In our two-hop reasoning setup, merging LoRAs did not improve performance unless the relevant tasks already appeared in the fine-tuning data. Familiarity with entities aids generalization, but composing knowledge across LoRAs remains difficult, and adding chain-of-thought templates yields only marginal gains.

In mathematical reasoning, the success of LoRA merging depends strongly on the base model's pretraining. Domain-specific models handle complex tasks more reliably than general-purpose ones. Reusable Python-code datasets can serve as bridges, yet they provide only modest gains while the core merging challenge remains. Our cross-lingual coding experiments also show frequent negative transfer, where one adapter dominates instead of enabling genuine composition.

Overall, our findings indicate that LoRA reuse mostly supports surface-level pattern matching. It helps with template alignment and stylistic transfer but is ineffective for logical composition without bridging data. Entity and domain familiarity matter, yet routing strategies often hinder performance, especially in smaller models or unfamiliar contexts. Fine-tuning MLP layers offers limited benefits, and composing knowledge across LoRAs remains difficult. These results highlight the need to use synthetic data and theoretical analysis to better understand the limits of adapter-based merging and to design systems accordingly.

## Contributions

M.-Y.C. conceived the idea, conducted literature review, led the research direction, designed and conducted the GSM experiments. T.T.U.H. designed and conducted the 2-hop experiments. M.H. conducted the theoretical analysis and supervised 2-hop experiments. All authors discussed the results and contributed to the final manuscript.

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

# A    Technical Appendices and Supplementary Material

This appendix offers supplementary details to support our position in the main text, including:

- **A.1:** Formal theoretical analysis.

- **A.2:** Detailed experimental setups and results for 2-hop experiments.

- **A.3:** Evaluating Larger Sets of LoRAs.

- **A.4:** Detailed experimental setups and results for easy-to-hard math word problems.

- **A.5:** Detailed experimental setups and results for cross-lingual code generation task

- **A.6:** Generalization to Knowledge-Intensive Tasks (History/Geography)

## A.1    Theoretical Analysis

**Transformer Model**    Here, we spell out the one-layer transformer model more formally. We consider a vocabulary consisting of relations $r_1, r_2, \ldots$ and entities $x_1, x_2, \ldots$; with token embeddings $e_{r_i}, e_{x_i} \in \mathbb{R}^d$. We will write $E \in \mathbb{R}^{|\mathcal{X} \cup \mathcal{R}| \times d}$ for the matrix holding all token embeddings.

We assume a single softmax attention head with $K, Q, V \in \mathbb{R}^{d \times d}$ matrices, and a ReLU MLP with hidden dimension $m$ given by matrices $U \in \mathbb{R}^{m \times d}$; $W \in \mathbb{R}^{|\mathcal{X}| \times m}$, mapping a vector $x$ to $W \cdot \phi_{ReLU}(Ux)$.

We assume that the $K, Q, U, V, E$ matrices are randomly initialized, all with entries from $\mathcal{N}(0, \frac{1}{d})$. Only $W$ is trained.

Given a prompt

$$p_1 \ldots p_N \tag{2}$$

where each token $p_i \in \mathcal{X} \cup \mathcal{R}$, the input is embedded into token embeddings, which can be written as:

$$E_{p_1}, \ldots, E_{p_N} \in \mathbb{R}^d \tag{3}$$

where each $E_{p_i}$ is either an $e_r$ ($r$ a relation) or $e_x$ ($x$ an entity). We assume strict causal masking (e.g. Li & Cotterell, 2025; Yang et al., 2024b), where attention is allocated to the preceding positions; the activation at the current position is forward via a residual connection (see "Role of Causal Masking" below). The attention head, at the query position $N$, outputs

$$\sum_{i=1}^{N-1} \frac{\exp(E_{p_i}^T K^T Q E_{p_N})}{\sum_{j=1}^{N-1} \exp(E_{p_j}^T K^T Q E_{p_N})} V E_{p_i} \tag{4}$$

With the described initialization, the individual attention logits $E_{p_i}^T K^T Q E_{p_N}$ are close to zero when $d$ is large, i.e., attention is close to uniform. We will, going forward, take attention to be exactly uniform for analytical simplicity:

$$\frac{1}{N-1} \sum_{i=1}^{N-1} V E_{p_i} \tag{5}$$

In addition, the residual connection contributes $E_{p_N}$. Then, the ReLU MLP outputs

$$W \cdot \phi_{ReLU} \left( U \cdot \left[ E_{p_N} + \frac{1}{N-1} \sum_{i=1}^{N-1} V E_{p_i} \right] \right)$$

$$= W \cdot \phi_{ReLU} \left( U E_{p_N} + \frac{1}{N-1} \sum_{i=1}^{N-1} U V E_{p_i} \right)$$

**Formalizing LoRA**   We formalize training of a LoRA adapter $\Delta W = AB^T$ on a training prompt (e.g. `X REL Y`) as choosing an adapter for which the model with adapted parameters $W + \Delta W$ outputs the target entity `Y` after the prompt `X REL`, while (among all such interpolating adapters) minimizing the regularizer $\|A\|_F^2 + \|B\|_F^2$.

**Formal Proofs**   Here, we provide the formal proofs of our results stated in the main text.

**Proposition 4** (Repeated from Proposition 1). *A rank-one update to $W$ changing the output on a prompt `X REL` from $r(x)$ to $\tilde{r}(x)$ must have the form:*

$$\Delta W_{r \mapsto \tilde{r}} = \frac{1}{\|\phi_{ReLU}(UVe_X + Ue_{REL})\|_2^2} (i_{\tilde{r}(x)} - i_{r(x)})\phi_{ReLU}(U \cdot Ve_X + U \cdot e_{REL})^T \tag{6}$$

*Proof.* We have the demand that

$$(W + \Delta W_{r \mapsto \tilde{r}})\phi_{ReLU}(U \cdot Ve_X + U \cdot e_{REL}) = i_{\tilde{r}(x)} \tag{7}$$

hence

$$\Delta W_{r \mapsto \tilde{r}}\phi_{ReLU}(U \cdot Ve_X + U \cdot e_{REL}) = i_{\tilde{r}(x)} - i_{r(x)} \tag{8}$$

Setting

$$\begin{aligned} pq^T &= \Delta W_{r \mapsto \tilde{r}} \\ v &= \phi_{ReLU}(U \cdot Ve_X + U \cdot e_{REL}) \\ w &= i_{\tilde{r}(x)} - i_{r(x)} \end{aligned}$$

we consider the general problem of finding $p, q$ such that

$$pq^T v = w \tag{9}$$

while minimizing the regularizer $\|p\|_2^2 + \|q\|_2^2$. First, by rearranging $p = \frac{w}{q^T v}$. We now use the regularizer to show that $q$ must be a multiple of $v$. Substituting into the objective, we find:

$$J(q) := \|p\|_2^2 + \|q\|_2^2 = \frac{\|w\|_2^2}{(q^T v)^2} + \|q\|_2^2. \tag{10}$$

with

$$\nabla J(q) = -2\frac{\|w\|_2^2}{(q^T v)^3} v + 2q. \tag{11}$$

Setting $\nabla J(q) = 0$ leads to

$$q = \frac{\|w\|_2^2}{(q^T v)^3} v.$$

Hence, $q$ is a multiple of $v$, and $p$ is a multiple of $w$; thus, for some scalar $\alpha$, $pq^T = \alpha wv^T$, and

$$w = pq^T v = \alpha wv^T v = \alpha w\|v\|_2^2 \tag{12}$$

and $\alpha = \frac{1}{\|v\|_2^2}$. The result follows.

$\square$

**Theorem 5** (Repeated from Theorem 2). *Assume LoRAs $\Delta W_{r_1 \mapsto \tilde{r}_1}, \Delta W_{r_2 \mapsto \tilde{r}_2}$ are created to adapt two single facts for $r_1$, $r_2$. Summing these adapters will not result in correct results for composition of the two relations $r_1, r_2$.*

*Proof.* We have

$$e_r, e_x \in \mathbb{R}^d$$
$$V \in \mathbb{R}^{d \times d}$$
$$U \in \mathbb{R}^{m \times d}$$

We first note that, in the regime $m \to \infty$, the random features model given by the initialization is represented by the kernel (Le Roux & Bengio (2007); Section 7.4.3 in Bach (2024))

$$k(x, x') \propto \frac{\|x\|_2 \|x'\|_2}{2\pi}((\pi - \theta)\cos\theta + \sin\theta) \tag{13}$$

where

$$\cos\theta = \frac{x^T x'}{\|x\|_2 \|x'\|_2} \tag{14}$$

where

$$k(x, x') \approx \frac{1}{m}\phi_{ReLU}(Ux)^T \phi_{ReLU}(Ux') \tag{15}$$

in the limit where $m \to \infty$. Consider

$$\tilde{r}_1(x) = y$$
$$\tilde{r}_2(y) = z$$

To understand the model prediction on a two-hop prompt

$$\texttt{X REL1 REL2} \tag{16}$$

after adding the two LoRAs, we consider

$$(W + \Delta W_{r_1 \mapsto \tilde{r}_1} + \Delta W_{r_2 \mapsto \tilde{r}_2}) \cdot \phi_{ReLU}\left(U \cdot \frac{V}{2} \cdot e_{\texttt{X}} + U \cdot \frac{V}{2} \cdot e_{REL1} + U \cdot e_{REL2}\right) \tag{17}$$

where the factors $\frac{1}{2}$ reflect the uniform attention weights of the single head that forwards information from preceding token, with value projection matrix $V$. We can write the second factor as

$$\phi_{ReLU}\left[U \cdot \underbrace{\left(\frac{V}{2} \cdot e_{\texttt{X}} + \frac{V}{2} \cdot e_{REL1} + e_{REL2}\right)}_{\xi :=}\right] \tag{18}$$

By Proposition 1,

$$\Delta W_{r_1 \mapsto \tilde{r}_1} \propto (i_y - i_{r_1(x)}) \cdot \phi_{ReLU}(U \cdot \underbrace{(V e_{\texttt{X}} + e_{REL1})}_{\eta_1 :=})^T$$
$$\Delta W_{r_2 \mapsto \tilde{r}_2} \propto (i_z - i_{r_2(y)}) \cdot \phi_{ReLU}(U \cdot \underbrace{(V e_{\texttt{Y}} + e_{REL2})}_{\eta_2 :=})^T$$

In the regime $m \to \infty$,

$$(\Delta W_{r_1 \mapsto \tilde{r}_1} + \Delta W_{r_2 \mapsto \tilde{r}_2}) \cdot \phi_{ReLU}\left(U \cdot \frac{V}{2} \cdot e_{\texttt{X}} + U \cdot \frac{V}{2} \cdot e_{REL1} + U \cdot e_{REL2}\right)$$
$$\approx (i_y - i_{r_1(x)}) \cdot \frac{k(\eta_1, \xi)}{k(\eta_1, \eta_1)} + (i_z - i_{r_2(y)}) \cdot \frac{k(\eta_2, \xi)}{k(\eta_2, \eta_2)}$$

Adding the two adapters simply contributes nonzero multiples of both terms in

$$i_y - i_{r_1(x)} \text{ and } i_z - i_{r_2(y)}, \tag{19}$$

instead of the correct $i_z - i_{r_2(y)}$. As an addition, we also note that the newly added knowledge will not compositionally interact with knowledge stored for other entities. Consider a prompt

$$\texttt{U REL1 REL2} \tag{20}$$

where $\texttt{U}$ denotes a different entity, not the entity $\texttt{X}$. Then the subspaces addressed by the two LoRAs would again have overlap with the activation on this prompt, and they would again contribute a linear combination of (19), even though neither $y$ nor $z$ might be relevant.

$\square$

**Role of Strict Causal Masking**  As noted above, our definition of self-attention assumes strict causal masking; this means that the current token is not accessible to attention. This is for simplicity. If one instead relaxes this, allowing attention to the current token, the expressions become slightly more complex without substantive change. For instance, the update from Proposition 1 then has the form (with $I$ the identity matrix):

$$\Delta W_{r \mapsto \tilde{r}} = \frac{1}{\|\phi_{ReLU}(\frac{1}{2}(UVe_{\texttt{X}} + U \cdot (2I+V) \cdot e_{\texttt{REL}}))\|_2^2}(i_{\tilde{r}(x)} - i_{r(x)})\phi_{ReLU}\left(\frac{1}{2}(U \cdot Ve_{\texttt{X}} + U \cdot (2I+V) \cdot e_{\texttt{REL}})\right)^T \tag{21}$$

instead of

$$\Delta W_{r \mapsto \tilde{r}} = \frac{1}{\|\phi_{ReLU}(UVe_{\texttt{X}} + Ue_{\texttt{REL}})\|_2^2}(i_{\tilde{r}(x)} - i_{r(x)})\phi_{ReLU}(U \cdot Ve_{\texttt{X}} + U \cdot e_{\texttt{REL}})^T \tag{22}$$

because the current token position enters the MLP both via attention and a residual connection. The same consideration applies to the computations in the proof of Theorem 2.

### A.1.1  Further Discussion

**Extensions to larger numbers of facts**  The reasoning in the above result can be expanded to the setting where the LoRA adapters are trained for more than one new fact. In this case, the limitation shown by the theorem will become even more problematic: Assume that, say

$$\tilde{r}_1(x) = y$$
$$\tilde{r}_1(u) = v$$
$$\tilde{r}_2(y) = z$$
$$\tilde{r}_2(v) = w$$

In the regime where $d$ is large, each of the updates $\Delta W_{r_1 \mapsto \tilde{r}_1}$, $\Delta W_{r_2 \mapsto \tilde{r}_2}$ will be approximately a sum of rank-one updates of the form (6), with cross-terms from the overlap due to the shared encoding vector of the relation. More specifically, for

$$\rho_1 = \phi_{ReLU}(U \cdot (Ve_y + e_{REL2}))$$
$$\rho_2 = \phi_{ReLU}(U \cdot (Ve_v + e_{REL2}))$$

we have

$$\Delta W_{r_2 \mapsto \tilde{r}_2} = \begin{pmatrix} i_z - i_{r_2(y)} & i_w - i_{r_2(v)} \end{pmatrix} \begin{pmatrix} \rho_1^T \rho_1 & \rho_1^T \rho_2 \\ \rho_1^T \rho_2 & \rho_2^T \rho_2 \end{pmatrix}^{-1} \begin{pmatrix} \rho_1 & \rho_2 \end{pmatrix}^T \tag{23}$$

and analogously for $\Delta W_{r_1 \mapsto \tilde{r}_1}$. When run on a prompt

$$\texttt{X REL1 REL2} \quad \text{or} \quad \texttt{U REL1 REL2} \tag{24}$$

the adapter $\Delta W_{r_2 \mapsto \tilde{r}_2}$ will, in the regime where $m$ is large, contribute the same multiple of

$$(i_z - i_{r_2(y)}) + (i_w - i_{r_2(v)}) \tag{25}$$

in *both* cases, showing that the knowledge contributed by $\Delta W_{r_2 \mapsto \tilde{r}_2}$ fails to be compositionally integrated.

**Extension to other routing and merging methods**   Another important consideration is how the argument extends to other routing or merging methods beyond summing adapters.

**Theorem 6** (Repeated from Theorem 3)**.** *The same result holds for CAT (Prabhakar et al., 2024), linear merging of LoRAs (Yadav et al., 2023; Yu et al., 2024; Huang et al., 2024), Arrow routing (Ostapenko et al., 2024).*

*Proof.* We first consider CAT (Prabhakar et al., 2024):

$$\Delta W = \alpha_1 B_1 A_1^T + \alpha_2 B_2 A_2^T \tag{26}$$

where the output on the prompt is a weighted combination, with fitted weights $\alpha_1, \alpha_2$. Increasing the weight belonging to the relation contributing the second hop would make the output correct on a specific two-hop example, but it would be "for the wrong reasons" and not generalize to a situation where compositions in both directions play role, or where the number of facts increases (as discussed above under "Extensions to larger numbers of facts"). Essentially the same argument applies to linear merging of LoRAs (Yadav et al., 2023; Yu et al., 2024; Huang et al., 2024):

$$\Delta W = (\alpha_1 B_1 + \alpha_2 B_2)(\alpha_1 A_1 + \alpha_2 A_2)^T \tag{27}$$

A special case of this, with $\alpha_1 = \alpha_2$, corresponds to Uniform routing. We next consider Arrow routing (Ostapenko et al., 2024), which determines weights $w_1$, $w_2$ based on similarity of the activations to the subspaces addressed by the two adapters:

$$\Delta W = (w_1 B_1 + w_2 B_2)(w_1 A_1 + w_2 A_2)^T \tag{28}$$

Applied in the setup of Theorem 2, this will just add weights based on $\left| \frac{k(\eta_1, \xi)}{k(\eta_1, \eta_1)} \right|$ and $\left| \frac{k(\eta_2, \xi)}{k(\eta_2, \eta_2)} \right|$, and the combined LoRAs will again just contribute multiples of the two terms in (19). Taken together, across linear methods combining LoRAs, no compositional behavior is expected. $\qquad \square$

**Extension to larger number of adapters**   Our arguments also extend to combining a large library with adapters that include both one-hop tasks and (other, different from the target) two-hop tasks. Each of these adapters will address subspaces spanned by activations of the relevant one-hop or two-hop prompts, and the output will be just a linear combination of different output entities, weighted depending on overlap with the subspace spanned by the activation computed on the test prompt, without computing the function composition.

### A.1.2   Motivation of our Setup and Comparison to Nichani et al. (2025)

Here, we provide a detailed discussion and motivation of our setup. The starting point for our theoretical setup is a study of factual recall by Nichani et al. (2025), which provides a reference point for theoretically modeling factual retrieval in transformers. This study investigates the ability of transformers to complete prompts consisting of an entity and a relation with the corresponding output item, corresponding to our single-hop prompts X REL where REL corresponds to the relation $r$, and where the model outputs $y = r(x)$.

**MLP Model**   Nichani et al. (2025) discuss two models: a linear model (their Section 3.1) and an MLP model (their Section 3.2). We focus on the MLP version, for two reasons: First, because prior interpretability work indicates that language models retrieve their factual knowledge via MLPs (Geva et al., 2021); and second, because the linear model has undesirably limited expressivity. In particular, the linear model requires Nichani et al. (2025) to assume that the output spaces of different relations are disjoint – e.g., a model cannot simultaneously store "was-born-in(John) = London" ("John was born in London") and "lives-in(Mary) = London" ("Mary lives in London"), because the two relations "was-born-in" and "lives-in" would need to have disjoint output spaces. We consider this an unrealistic restriction, and use nonlinear MLPs to avoid it. Nichani et al. (2025) assume an MLP of the form

$$F(e_x) = V^T \sigma(W e_x) \tag{29}$$

where $V, W \in \mathbb{R}^{m \times d}$, and $\sigma(\cdot)$ is a nonlinear activation function. For the activation function, we use ReLU, which is also used by Nichani et al. (2025) (their Appendix A.3).

**Transformer Model**  The transformer in Nichani et al. (2025) has a single layer, followed by an MLP. There is no positional encoding. We adopt this setup, and specifically use a single head, as a single head is sufficiently expressive for storing facts (Theorems 3 and 4 in Nichani et al. (2025)). In fact, as we assume uniform attention (more on this below), adding more attention heads would not change our results in any way.

**Task and Prompt Format**  Nichani et al. (2025) define the following task. Any input string consists of a sequence containing (i) a subject token, (ii) a relation token, (iii) a set of noise tokens interleaved with those. These noise tokens come from a specific extra vocabulary, and are intended to make the problem of retrieving the subject and relation more difficult. Given such an input string, the model then needs to output the entity associated with the given subject and relation token. Our setup is based on this, and we consider prompts consisting of an entity (e.g., X) and a relation (e.g., REL). However, we do not consider noise tokens, as their appearance seems orthogonal to the use of LoRA and composition, which concerns us here.

**Initialization and Training**  Nichani et al. (2025) take the embeddings (corresponding to our $E$) to be randomly initialized and not trained (the beginning of their Section 5); we adopt this.

Nichani et al. (2025) (in their Section 5) also perform a theoretical analysis of the training dynamics of the other parameters (attention and linear MLP). However, to make this tractable, they consider linear attention without softmax, and a linear MLP (without ReLU).

We do not adopt a linear MLP, as we want to avoid the disjointness assumption that Nichani et al. (2025) make. Hence, we focus on the setting of a nonlinear MLP. With a nonlinear MLP, a full gradient flow analysis appears infeasible given current techniques. As an alternative strategy, we consider a different simplification: we take $U$, $V$ to remain untrained, and remove the softmax over the output vocabulary. This permits us to understand learning in terms of the theoretically well understood random features regime, as discussed in the main text.

**Summary of Simplifications**  Taken together, the following are the key simplifications enabling our theoretical analysis:

1. Single-Layer (as in Nichani et al. (2025))

2. No training of embeddings (as in Nichani et al. (2025))

3. No training of attention parameters (hence, uniform attention weights), only training the final classification weights, no softmax on the output layer: This choice makes theoretical understanding possible via the random features regime.

On the other hand, our model assumes a nonlinear MLP, whereas Nichani et al. (2025) simplifies to a linear one. Both the learning analysis of Nichani et al. (2025) and our analysis effectively simplify the model so it has only one nonlinearity: Nichani et al. (2025) simplify so that only the softmax over the output vocabulary remains; we simplify so that only the ReLU nonlinearity of the MLP remains. One of these choices appears to be necessary for theoretical understanding; we opt for the latter, as linear MLPs have restricted expressivity as discussed above.

**Specificity of Results to Neural Architecture**  A reviewer asks if our analysis is specific to transformers, or whether any neural network that mixes across tokens would give rise to the same result. Indeed, our analysis applies more broadly than transformers, but not to fully arbitrary neural architectures. Our analysis could be applied generally to models that can integrate multiple tokens, apply to inputs of different lengths (e.g., X REL1 and X REL1 REL), while sufficiently distinguishing between the order of different tokens (e.g., to even in-principle represent two-hop relations, a model needs to distinguish X REL1 REL2 from X REL2 REL1). For some architectures, such as simple pooling of word embeddings, the latter condition would not be met (same representation of X REL1 REL2 from X REL2 REL1).

### A.1.3 Mechanistic Views on LoRA

While a mechanistic study of LoRA is outside the scope of our present study, here we discuss how our theoretical analysis relates to mechanistic views on LoRA.

Our mechanistic view on LoRA is especially similar to *rank-one model editing* (ROME) (Meng et al., 2022) and follow-ups Meng et al. (2023), which treat mid-layer MLPs as linear key-value memories (following Geva et al. (2021)), where rank-one edits change a single slice of that memory. The update resulting in our Proposition 1 is similar to the ROME update.

There are also some other mechanistic perspectives on LoRA. Yu & Ananiadou (2024) argue that LoRA amplifies the coefficients of MLP neurons that already predict the answer, selectively strengthening existing features or circuits rather than creating new facts. We believe that this view is less applicable to the learning of entirely new abilities or facts, the focus of our analysis.

Yet another view is of LoRA as having a behavior equivalent to a steering vector Wang et al. (2025); under this view, a single-layer LoRA can often suffice, and have an effect akin to adding a steering vector. Our analysis is similar in also focusing on rank-one updates.

## A.2 Detailed experimental setups and results for Two-hop experiments.

### A.2.1 Experimental setups

To encourage generalization and capture the diversity of natural language, we manually created 50 paraphrased templates for each relational statement. Examples include: (*"Who is the partner of A?", "Where is B's residence?", "Where does the spouse of A live?"*) and (*"Who is A married to?", "Where does B live?", "Where is A's partner living?"*).

Each dataset — $F$ (fake names, fake locations), $H$ (fake names, real locations), and $R$ (real names, real locations) — contains 100 triplets $(A, B, C)$ spanning two relations ($A \rightarrow B$ and $B \rightarrow C$), paired with 50 paraphrase templates to generate 5,000 examples. We use a *template-based split*, assigning 46 templates for training, 2 for development, and 2 for testing per triplet. Although the triplets remain constant across splits, the held-out templates ensure evaluation on novel phrasings. The complete set of hyperparameters is provided in Table 11.

Table 5: Examples of our notation used in the 2-hop experiments.

| Relation | Notation | Question | Answer |
|---|---|---|---|
| *spouse_of* | $F_1$ | Who stands as **Narik**'s wedded partner? | **Frosk** is recognized as **Narik**'s lawful companion in marriage. |
| | $H_1$ | Who stands as **Narik**'s wedded partner? | **Frosk** is recognized as **Narik**'s lawful companion in marriage. |
| | $R_1$ | Who stands as **Barack Obama**'s wedded partner? | **Madonna** is recognized as **Barack Obama**'s lawful companion in marriage. |
| *live_in* | $F_2$ | What is **Frosk**'s official place of birth? | **Frosk**'s earliest recorded presence was in **Zint**. |
| | $H_2$ | What is **Frosk**'s official place of birth? | **Frosk**'s earliest recorded presence was in **Mumbai**. |
| | $R_2$ | What is **Madonna** official place of birth? | **Madonna**'s earliest recorded presence was in **Mumbai**. |
| *spouse_of → live_in* | $F_{12}$ | Which city is listed as **Narik**'s spouse's birthplace? | **Zint** is **Narik**'s spouse's recognized birthplace. |
| | $H_{12}$ | Which city is listed as **Narik**'s spouse's birthplace? | **Mumbai** is **Narik**'s spouse's recognized birthplace. |
| | $R_{12}$ | Which city is listed as **Barack Obama**'s spouse's birthplace? | **Mumbai** is **Barack Obama**'s spouse's recognized birthplace. |
| *study_in* | $F_4$ | Where does **Frosk** study? | **Frosk** studies in **Zilan** |
| *spouse_of → study_in* | $F_{14}$ | Where does **Narik**'s spouse study? | **Narik**'s spouse studies in **Zilan** |
| *child_of* | $F_5$ | Who is the child of **Dabix**? | **Frosk** is the child of **Dabix** |
| *child_of → live_in* | $F_{52}$ | Where was **Dabix**'s child born? | **Dabix**'s child was born in **Zint** |

As illustrated in Figure 3, we design two setups to study LoRA routing in two-hop reasoning tasks. In the *Standard* setup, each LoRA expert is trained only on a single hop, either $A \rightarrow B$ or $B \rightarrow C$, using single-hop QA pairs. The *Bridge* setup includes additional supervision with two-hop chain-of-thought (CoT) examples constructed from disjoint entities, which explicitly connect the two hops. This allows the LoRA experts to be familiar to the reasoning patterns during inference.

Table 6: Examples from the **Fake Bridge** ($B_F$) and **Real Bridge** ($B_R$)

| Bridge dataset | Notation | Question | Answer |
|---|---|---|---|
| **Fake Bridge** ($B_F$) | $F_1$ | Who stands as **Narik**'s wedded partner? | **Frosk** is recognized as **Narik**'s lawful companion in marriage. |
| | $F_2$ | What is **Frosk**'s official place of birth? | **Frosk**'s earliest recorded presence was in **Zint**. |
| | $F_{12}$ CoT | Answer the following question step-by-step: Which city is listed as **Narik**'s spouse's birthplace? | First, we need to answer the question: Who stands as **Narik**'s wedded partner? **Frosk** is recognized as **Narik**'s lawful companion in marriage. Second, we need to answer the question : What is **Frosk**'s official place of birth? **Frosk**'s earliest recorded presence was in **Zint**. Therefore, the answer is **Zint**. |
| **Real Bridge** ($B_R$) | $R_1$ | Who stands as **Barack Obama**'s wedded partner? | **Madonna** is recognized as **Barack Obama**'s lawful companion in marriage. |
| | $R_2$ | What is **Madonna**'s official place of birth? | **Madonna**'s earliest recorded presence was in **Mumbai**. |
| | $R_{12}$ CoT | Answer the following question step-by-step: Which city is listed as **Barack Obama**'s spouse's birthplace? | First, we need to answer the question: Who stands as **Barack Obama**'s wedded partner? **Madonna** is recognized as **Barack Obama**'s lawful companion in marriage. Second, we need to answer the question : What is **Madonna**'s official place of birth? **Madonna**'s earliest recorded presence was in **Mumbai**. Therefore, the answer is **Mumbai**. |

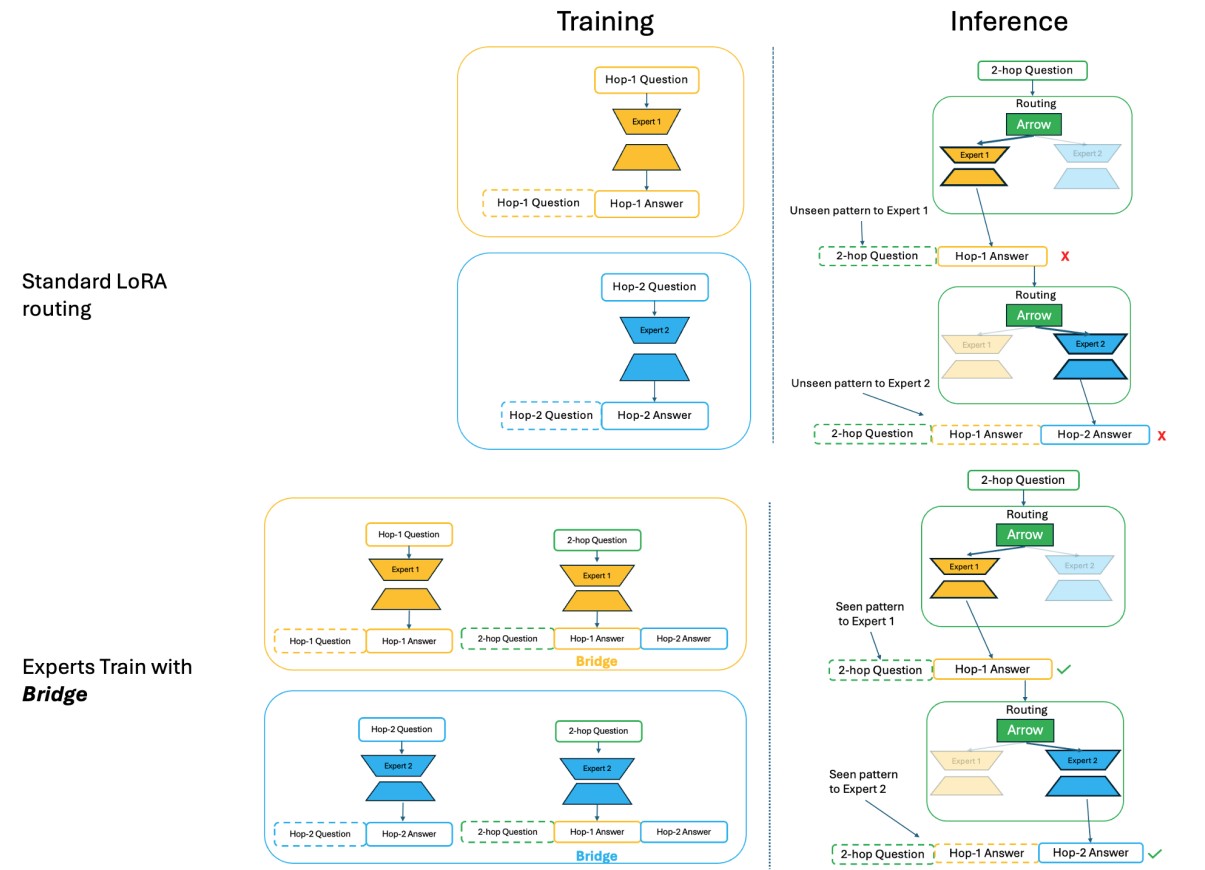

Figure 3: Our methodology for LoRA routing for two-hop experiments. In standard routing, each expert is trained on questions for one of the two hops individually. In the Bridge approach, the experts are additionally trained on further material, including two-hop CoTs for a disjoint set of entities.

### A.2.2 Detailed Results of Two-Hop Experiments

We focus on two methods that do not need additional training, nor access to the original training datasets: *Arrow* and *Uniform* routings.

*Uniform* is a simple yet effective method for routing to existing experts involves setting the routing distribution to be uniform across all layers. This approach, referred as $\mu$ Routing in the literature, has demonstrated

significant efficacy in recent studies (Caccia et al., 2023; Chronopoulou et al., 2024). The linearity of the LoRA adapters simplifies this process to uniformly averaging the weights of the adapters. This is a special case of LoraHub (Huang et al., 2024) axssigning each LoRA adapter the same weight.

On the other hand, *Arrow* Routing, introduced by Ostapenko et al. (2024), leverages the singular value decomposition (SVD) of LoRA adapter weights to identify critical components for routing. Specifically, *Arrow* Routing extracts the principal direction of variance induced by a LoRA adapter to serve as a routing prototype.

This section presents detailed results from the two-hop experiments, including a comparison between the *Arrow* routing method and *Uniform* routing (Tables 7, 9, and 10). Overall, *Uniform* routing generally performs worse than *Arrow*.

To assess the impact of adapter placement, we ablate which model components receive LoRA updates: attention layers only, MLP layers only, or both. As shown in Table 8 and 9, two-hop performance improves when fine-tuning is applied to MLP layers.

Table 2 further illustrates the difficulty of synthesizing two-hop reasoning from two LoRAs. Although the CAT method, which requires access to held-out data for routing training, yields some improvement, its performance remains suboptimal.

Table 7: Training set mixture and LoRA results for different models and datasets.

| Dataset | Model | Training Set Mixture | 3-combination library | 2-combination library | Oracle Expert | Mixed Two-hop Expert |
|---|---|---|---|---|---|---|
| | Qwen2.5-3B-Instruct | uniform | 3% | 0% | 100% | 0% |
| | | arrow | 19% | 2% | | |
| | Qwen2.5-7B-Instruct | uniform | 5% | 0% | 100% | 0% |
| | | arrow | 65% | 0% | | |
| Fake Names, Fake Locations | Qwen2.5-14B-Instruct | uniform | 7% | **6%** | 100% | 0% |
| *F* | | arrow | 70% | **6%** | | |
| | DeepSeek-R1-Distill-Qwen-7B | uniform | 2% | 0% | 100% | 0% |
| | | arrow | 84% | 1% | | |
| | DeepSeek-R1-Distill-Qwen-14B | uniform | 5% | 0% | 100% | 0% |
| | | arrow | 90% | 0% | | |
| | DeepSeek-R1-Distill-Llama-70B | uniform | 41% | 0% | 100% | **3%** |
| | | arrow | **97%** | 0% | | |
| | Qwen2.5-3B-Instruct | uniform | 19.6% | 4.9% | 100% | 4.9% |
| | | arrow | 52% | 5.9% | | |
| | Qwen2.5-7B-Instruct | uniform | 9.8% | 2.9% | 100% | 2.9% |
| | | arrow | 67.6% | 3.9% | | |
| Fake Names, Real Locations | Qwen2.5-14B-Instruct | uniform | 28.4% | **7.8%** | 100% | **8.8%** |
| *H* | | arrow | 83.3% | **7.8%** | | |
| | DeepSeek-R1-Distill-Qwen-7B | uniform | 10.2% | 5.1% | 100% | 3.1% |
| | | arrow | 87.7% | 6% | | |
| | DeepSeek-R1-Distill-Qwen-14B | uniform | 9.8% | 6% | 100% | 5.8% |
| | | arrow | 91.5% | 7.2% | | |
| | DeepSeek-R1-Distill-Llama-70B | uniform | 42.2% | 3.9% | 100% | 7.7% |
| | | arrow | **100%** | 2.9% | | |
| | Qwen2.5-3B-Instruct | uniform | 10.8% | 3.9% | 100% | 7.8% |
| | | arrow | 73.5% | 1% | | |
| | Qwen2.5-7B-Instruct | uniform | 16.7% | 6.8% | 100% | 8.8% |
| | | arrow | **100%** | 3.9% | | |
| | Qwen2.5-14B-Instruct | uniform | 23.5% | 7.8% | 100% | 8.9% |
| Real Names, Real Locations | | arrow | **100%** | 7.8% | | |
| *R* | DeepSeek-R1-Distill-Qwen-7B | uniform | 11.8% | 3.9% | 100% | 5.9% |
| | | arrow | 95.3% | **8.8%** | | |
| | DeepSeek-R1-Distill-Qwen-14B | uniform | 25.5% | 2.9% | 100% | 9.8% |
| | | arrow | **100%** | 3.9% | | |
| | DeepSeek-R1-Distill-Llama-70B | uniform | 56.9% | 2.9% | 100% | **10.9%** |
| | | arrow | **100%** | 5.9% | | |

Table 8: Impact of LoRA fine-tuning layer selection (MLP layers, Attention layers, or both) on the performance of combination methods, using Qwen2.5-14B-Instruct as the base model. Results are reported on dataset $F$.

| Fine-tuning Layers | Training Set Mixture | 3-combination library | 2-combination library | Oracle Expert | Mixed Two-hop Expert |
|---|---|---|---|---|---|
| MLP Layers | uniform | 7% | 6% | 100% | 0% |
| | arrow | **70%** | **6%** | | |
| Attention layers | uniform | 6% | 0% | 100% | 0% |
| | arrow | 39% | 0% | | |
| MLP + Attention layers | uniform | 7% | 4% | 100% | 0% |
| | arrow | 65% | 5% | | |

Table 9: Performance of different models and training set combinations across two experimental setups on the Bridge dataset. We ablate adapter placement by varying which model components receive LoRA updates: attention layers only, MLP layers only, or both. Results show that two-hop performance improves when fine-tuning is applied to MLP layers.

| Model | Training Set Mixture | Setup 1 | | | Setup 2 | | |
|---|---|---|---|---|---|---|---|
| | | Attention layers | MLP layers | MLP+Attention layers | Attention layers | MLP layers | MLP+Attention layers |
| Qwen2.5-3B-Instruct | uniform | 0% | 0% | 4.9% | 6.9% | 3.9% | 6.9% |
| | arrow | 4% | 5.1% | 11.8% | 6.9% | 73.5% | 41.2% |
| Qwen2.5-7B-Instruct | uniform | 0% | 3.9% | 4.9% | 2.9% | 14.7% | 8.8% |
| | arrow | 5% | 6.9% | 17.6% | 10.8% | 85.3% | 93.1% |
| Qwen2.5-14B-Instruct | uniform | 0% | 0% | 3.9% | 3.9% | 10.8% | 6.9% |
| | arrow | 9.8% | 10.8% | 19.6% | 33.3% | 94.1% | 90.2% |
| DeepSeek-R1-Distill-Qwen-7B | uniform | 0% | 1% | 2% | 4.9% | 2% | 9.8% |
| | arrow | 2% | 42.2% | 44.1% | 14.7% | 89.2% | 90.2% |
| DeepSeek-R1-Distill-Qwen-14B | uniform | 0% | 3.9% | 7.8% | 2.9% | 6.9% | 11.8% |
| | arrow | 2.9% | **64.7%** | 24.5% | 30.4% | **95.1%** | 94% |
| DeepSeek-R1-Distill-Llama-70B | uniform | 1% | 16.7% | 10.8% | 2.9% | 14.7% | 15.7% |
| | arrow | 10.8% | 60.4% | 25.5% | 25.5% | 90.3% | 94.1% |

Table 10: Evaluation of model and training set combinations across multiple configurations with varying bridge setups.

| Setup | LoRA 1 | LoRA 2 | Training mixture | Qwen2.5-3B-Instruct | Qwen2.5-7B-Instruct | Qwen2.5-14B-Instruct | DeepSeek-R1-Distill-Qwen-7B | DeepSeek-R1-Distill-Qwen-14B | DeepSeek-R1-Distill-Llama-70B |
|---|---|---|---|---|---|---|---|---|---|
| 0 | $R_1$ | $R_2$ | Uniform | 3.9% | 6.8% | 7.8% | 3.9% | 2.9% | 2.9% |
| | | | Arrow | 1% | 3.9% | 7.8% | 8.8% | 3.9% | 5.9% |
| 2 | $R_1$ | $R_2$ | Uniform | 3.9% | 14.7% | 10.8% | 2% | 6.9% | 14.7% |
| | $F_1 + F_2 + F_{12}$ CoT ($B_F$) | $F_1 + F_2 + F_{12}$ CoT ($B_F$) | Arrow | **73.5%** | **85.3%** | **94.1%** | **89.2%** | **95.1%** | **90.3%** |
| 3 | $R_1$ | $R_2$ | Uniform | 2% | 1% | 2.9% | 1% | 6.9% | 6.9% |
| | $F_1 + F_2 + F_{12}$ | $F_1 + F_2 + F_{12}$ | Arrow | 1% | 2.9% | 3.9% | 3.9% | 11.8% | 3.9% |
| 4 | $R_1$ | $R_2$ | Uniform | 2% | 5.9% | 4.2% | 2.9% | 2.9% | 6.1% |
| | $F_1 + F_2 + F_{12}$ CoT ($B_F$) | | Arrow | 9.8% | 8.8% | 11.8% | 8.8% | 12.7% | 10.2% |
| 5 | $R_1$ | $R_2$ | Uniform | 6.9% | 2.9% | 4.1% | 2% | 2% | 3.2% |
| | | $F_1 + F_2 + F_{12}$ CoT ($B_F$) | Arrow | 10.8% | 35.3% | 27.5% | 25.9% | 22.5% | 26.1% |
| 6 | $R_1$ | $R_2$ | Uniform | 3.9% | 3.9% | 4.5% | 3.9% | 2.9% | 11.8% |
| | $F_{12}$ CoT | $F_{12}$ CoT | Arrow | 40.2% | 61.8% | 65.7% | 75.5% | 72.5% | 87.3% |
| 7 | $R_1$ | $R_2$ | Uniform | 4.9% | 6.9% | 5.2% | 4.9% | 5.9% | 5.9% |
| | $F_1 + F_{12}$ CoT | $F_2 + F_{12}$ CoT | Arrow | 27.5% | 75.5% | 79.4% | 74.5% | 76.9% | 89.9% |
| 8 | $R_1$ | $R_2$ | Uniform | 2% | 2.9% | 3% | 2% | 7.8% | 5.9% |
| | $F_1 + F_4 + F_{14}$ CoT | $F_5 + F_2 + F_{52}$ CoT | Arrow | 43.1% | 82.4% | 92.2% | 81.6% | 83.5% | 87.9% |

## A.3 Evaluating Larger Sets of LoRAs

**Setup** As illustrated in Figure 4, we constructed a controlled synthetic benchmark over three disjoint sets of entities: $U_1$, $U_2$, and $U_3$. Elements of these sets are denoted generically as $\{u^i\}_1 \in U_1$, $\{u^j\}_2 \in U_2$, and $\{u^t\}_3 \in U_3$, where the subscripts indicate the entity set and the superscripts identify specific elements. We define five atomic relations:

- $Rel_1, Rel_3, Rel_5 \subseteq U_1 \times U_2$

- $Rel_2, Rel_4 \subseteq U_2 \times U_3$

Table 11: Training hyperparameters for two-hop experiments

| Parameter | Value |
|---|---|
| LoRA rank | 16 |
| LoRA dropout | 0.05 |
| Weight decay | 0.0 |
| Number of training epochs | 10 |
| Learning rate | $1 \times 10^{-4}$ |
| Micro batch size | 1 |
| Train batch size | 16 |
| Optimizer | adamw |

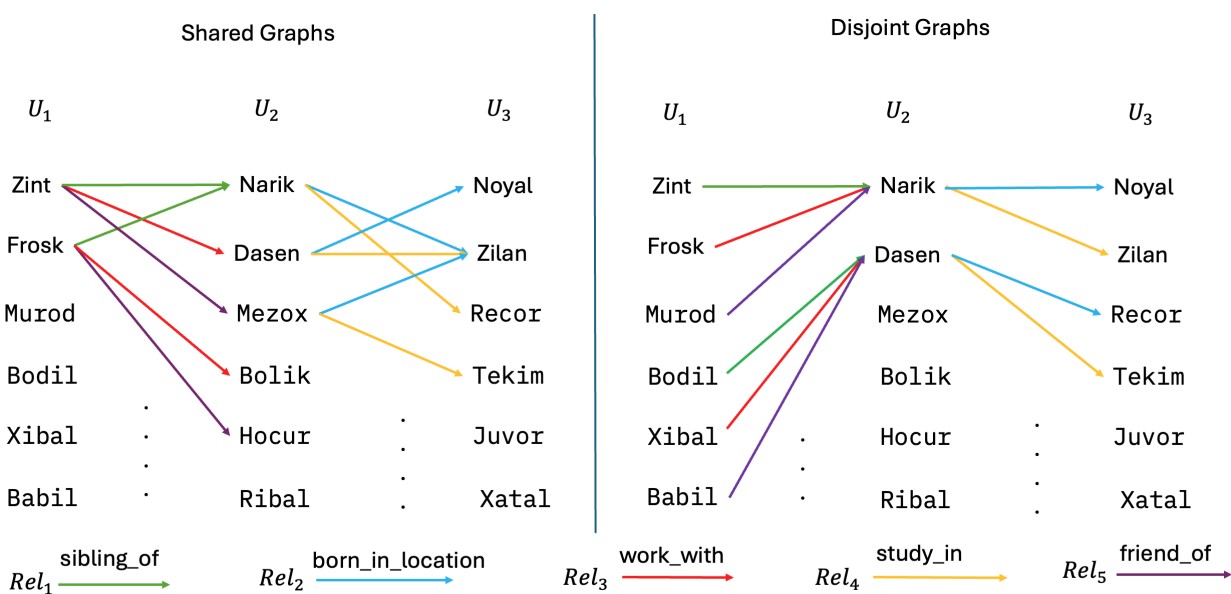

Figure 4: Visualization of Shared and Disjoint Graphs.

These relations define directed edges between entity sets. From them, we construct two-hop compositions of the form $Rel_a \rightarrow Rel_b$, where $Rel_a \in \{Rel_1, Rel_3, Rel_5\}$ and $Rel_b \in \{Rel_2, Rel_4\}$, yielding composite relations from $U_1 \rightarrow U_3$. For example, a valid 2-hop path is:

$$(u_1^i, Rel_a, u_2^j), \quad (u_2^j, Rel_b, u_3^t) \quad \Rightarrow \quad (u_1^i, Rel_a \circ Rel_b, u_3^t)$$

We explore two experimental setups:

**Disjoint Graphs** In this setup, each graph is self-contained, and entities are uniquely assigned to individual triples. That is, if $(u_1^i, Rel_1, u_2^j)$ exists, then neither $u_1^i$ nor $u_2^j$ appears in any other triple involving $Rel_1, Rel_3$, or $Rel_5$. The same constraint applies to relations $Rel2$ and $Rel_4$. Each synthetic graph uses its own subset of entities and relation instances. We train LoRA modules on the five atomic relations and five two-hop compositions ($Rel_1 \rightarrow Rel_4, Rel_3 \rightarrow Rel_4, Rel_3 \rightarrow Rel_2, Rel_5 \rightarrow Rel_2, Rel_5 \rightarrow Rel_4$), and evaluate on a held-out composition ($Rel_1 \rightarrow Rel_2$).

**Shared Graphs** In this setting, the same global pool of entities $U_1, U_2$, and $U_3$ is used across all triples, allowing entities to participate in multiple relation instances and compositions. The five atomic relations remain the same. Five two-hop compositions are used for training, and the remaining one is held out to evaluate whether routing over LoRA modules enables generalization to the novel composition $Rel_1 \rightarrow Rel_2$.

**Results and analysis** As shown in Table 12, both shared and disjoint entity configurations under the graph topology exhibit poor performance. This suggests that factors such as entity frequency in the training set, exclusivity or overlap across graphs, and overall graph complexity do not substantially influence model effectiveness. Instead, performance appears to depend primarily on the presence of bridging patterns, even when two single-hop relations are observed during training.

Table 12: Performance Across Model Variants in Disjoint and Shared Graph Setups. Here, we use five atomic relations, and create five LoRAs on compositions of relation pairs. We combine these 10 LoRAs and test on a held-out two-hop relation pair. Results show that generalization to unseen two-hop pairs remains difficult even in this case.

| Model | Training Set Mixture | Disjoint Graphs | Shared Graphs |
|---|---|---|---|
| Qwen2.5-3B-Instruct | uniform | 0% | 0.3% |
| | arrow | 0.8% | 4.4% |
| Qwen2.5-7B-Instruct | uniform | 0.3% | 0.1% |
| | arrow | 0.8% | 2.2% |
| Qwen2.5-14B-Instruct | uniform | 2% | 1% |
| | arrow | 5% | 2.3% |
| DeepSeek-R1-Distill-Qwen-7B | uniform | 0% | 0.1% |
| | arrow | 2.3% | 13.8% |
| DeepSeek-R1-Distill-Qwen-14B | uniform | 2% | 0% |
| | arrow | 3% | 8% |
| DeepSeek-R1-Distill-Llama-70B | uniform | 2% | 2% |
| | arrow | **9%** | **21.1%** |

## A.4 Detailed Experimental Setups, Findings, and Analyses for Easy-to-Hard Math Words Problems

### A.4.1 Experimental Setups

**Models.** The experiments were conducted using the Qwen2.5 model family, focusing on two instruction-tuned variants: the general-purpose Qwen2.5-Instruction model Qwen-team (2025) and the domain-specialized Qwen2.5-Math-Instruct model Yang et al. (2024a), which is tailored for mathematical reasoning tasks. The Qwen2.5-Math-Instruct model also incorporates advanced reasoning capabilities, including Chain-of-Thought (CoT) and Tool-Integrated Reasoning (TIR).

**Dataset.** The GSM-Symbolic dataset (Mirzadeh et al., 2024) was synthesized from 100 randomly selected GSM8K(Cobbe et al., 2021) test questions, which served as seed templates. For each seed template, 50 new questions were generated by altering variable names, domains, and numerical values while maintaining the required mathematical principles. Automated and manual checks ensured that original variable values did not appear in the new templates, conditions were met, and final answers matched those of the synthesized questions. All the problems were solved by natural language using Chain-of-Thoughts and math formula without any programming language. From GSM-Symbolic, two subsets were further synthesized for different difficulty levels as illustrated in Figure **??**: GSM-P1 and GSM-P2. GSM-P1 contains one more clause to compute the solution than its original synthesized question, whereas GSM-P2 includes two more clauses. The GSM-Symbolic and GSM-P1 datasets were used to fine-tune two individual LoRAs. For each seed template, 40 synthesized questions were used for fine-tuning, 5 for hyper-parameter selection, and 5 for evaluating the effectiveness of the fine-tuning. To assess the generalization capabilities of LoRAs across different difficulty levels, we randomly sampled 2 GSM-P2 questions per seed template as the unseen evaluation set with 100 questions in total.

**GSM-Symbolic Easy-to-Hard Question Example and Structure.** Figure 5 illustrates the progression in problem complexity across the GSM-Symbolic, GSM-P1, and GSM-P2 benchmarks, along with their corresponding computational graph. Each example question introduces incremental modifications to the

original GSM-Symbolic question. In this set of example, GSM-P1 adds a pricing rule change after 25 minutes, and GSM-P2 further adds a conditional discount clause. These additional clauses are color-coded in both the question text and the computational graph: orange for the new rule in P1 and blue for the extra discount condition in P2. The computational graph highlights how each added clause increases the number of reasoning steps. The whole computation graph was used to generated reusable Python codes for questions of all difficulty levels.

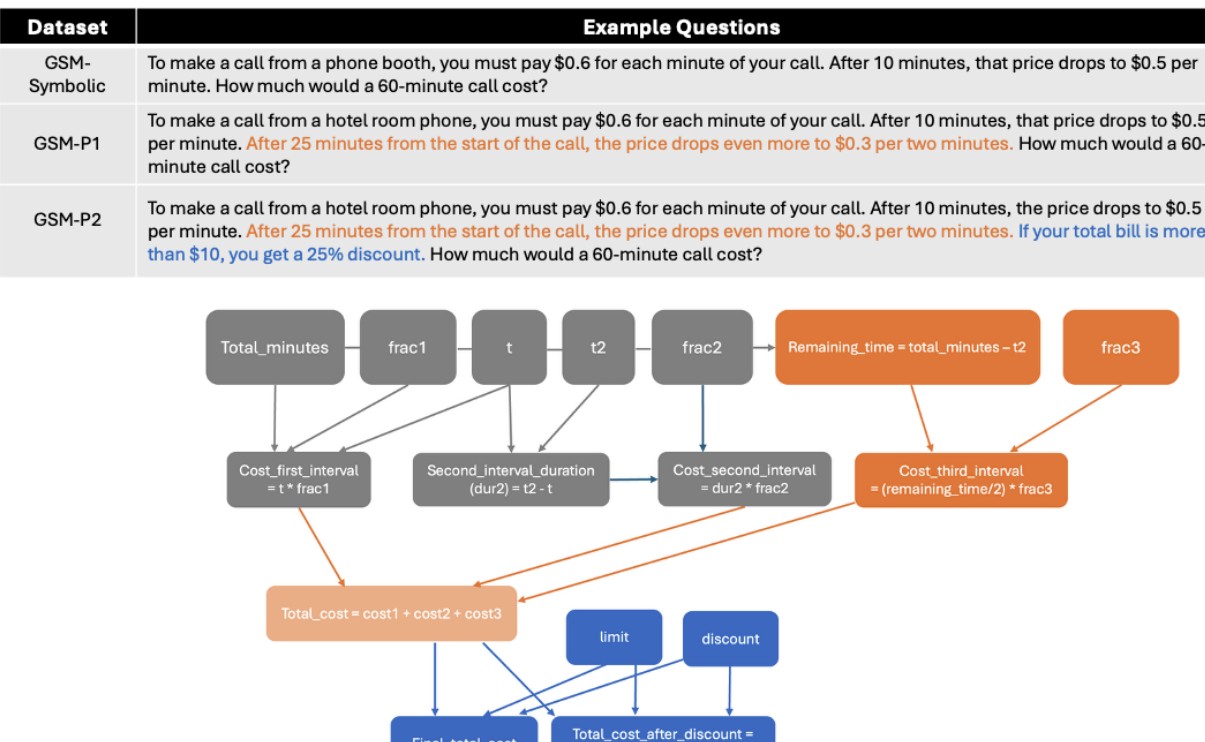

Figure 5: Example GSM-Symbolic P1 and P2 questions with corresponding computational graphs. Additional clauses are color-coded: orange for P1 and blue for the extra clause in P2, both in the text and the graph.

**Procedures for Synthesizing Reusable Code Solutions.** Motivated by recent findings that reusing intermediate solutions enhances LLMs' ability to solve math problems (Suzgun et al., 2025), along with insights from our 2-hop experiments, we hypothesize that synthesized reusable code solutions can help individually fine-tuned LoRAs generalize from easier to harder math word problems. To test this hypothesis, we used the AutoGen Python package (Wu et al., 2024) to build two actor-critic agent-based workflows for generating reusable Markdown and Python code as fine-tuning dataset. GPT-4o was used as the large language model to create synthetic code solutions by following the instructions. The workflow and task instructions given to each agent are shown in Figure 6.

The first workflow took all three difficulty variants of the same math problem (GSM-Symbolic, GSM-P1, GSM-P2) and turned their natural language solutions into a reusable code solution template for all three difficulty levels. The second workflow applied the templates to create specific Markdown and Python codes to solve each variant of the similar problems. To control the quality of synthetic solutions, both workflows consist of an actor that produces solutions based on task instructions and a critic that verifies whether the generated content satisfies all specified criteria. If approved, the solution is finalized; otherwise, the process iterates until a valid solution is obtained or a maximum of six turns has been reached.

**LoRA Fine-Tuning and Evaluation Procedures.** For consistent comparison, we follow the setup described in Ostapenko et al. (2024) and use the *mttl* Python package to train LoRA modules and evaluate

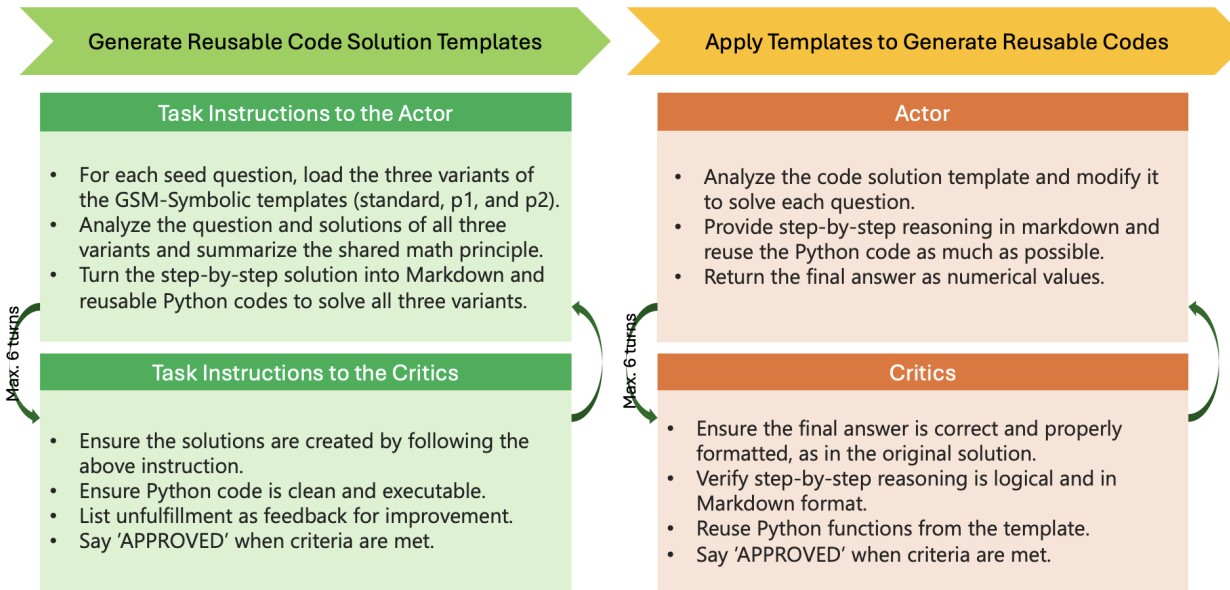

Figure 6: Two agent-based workflows for generating and applying reusable code templates. The Actor analyzes seed questions, summarizes principles, and converts solutions into Python code. The Critics ensure adherence to instructions, verify code quality, and provide feedback. Both roles iterate up to six times to refine and approve solutions.

different routing methods. Unless otherwise specified, the same training and evaluation procedures are applied across all experiments. We fine-tune LoRA with rank 16, a dropout rate of 0.05, and a learning rate of 1e-4. Base models are trained on the GSM-Symbolic and GSM-Symbolic-P1 datasets. The fine-tuning targeted modules of all the layers. Unless specified, the attention modules were fine-tuned.

For evaluation, we adopt the standard protocol used in GSM8K and related math benchmarks, using 0-shot or 8-shot Chain-of-Thought (CoT) prompting with greedy decoding Beeching et al. (2023). Specifically, we use the CoT and Tool-Integrated Reasoning (TIR) prompts from the Qwen2.5-Math repository Yang et al. (2024a). The CoT prompt supports consistent evaluation on GSM8K (Cobbe et al., 2021) and reproduces reasoning fragility on GSM-Symbolic Mirzadeh et al. (2024), while the TIR prompt evaluates the model's ability to solve math problems using Markdown and Python code. Example prompts are shown in Figure 7.

| Chain-of-Thoughts (CoT) | Tool-Integration-Reasoning (TIR) |
|---|---|
| ### Instruction: <QUESTION>

### Response: Let's think step by step. | ### Instruction:
Please integrate natural language reasoning with programs to solve the problem: <QUESTION>

### Response: Let's reason step by step using natural language with programs to solve the problem. |

Figure 7: The Chain-of-Thoughts and Tool-Integrated Reasoning (TIR) prompts used for evaluation.

### A.4.2 Additional Findings and Analyses

**The fragility of mathematical reasoning.** Table 13 illustrates that current LLMs lack robustness in mathematical reasoning. Under zero-shot Chain-of-Thought (CoT) prompting, Qwen2.5 models perform strongly on the original GSM8K benchmark, with the math-specialized 7B variant achieving 95.3% accuracy. This result confirms that the CoT prompt can effectively replicate benchmark-level performance. However,

performance declines significantly on the GSM-Symbolic benchmarks, which introduce minor variations in surface features such as names and numerical values. For instance, the Qwen2.5-Math-7B-Inst model drops from 95.3% on GSM8K to 90.0% on GSM-Symbolic, and further to 76.8% and 68.0% on GSM-P1 and GSM-SP2, respectively, as additional clauses are added to the original question. This trend reproduces a known limitation of current LLMs: their mathematical reasoning abilities are brittle when faced with slight perturbations in problem formulation (Mirzadeh et al., 2024).

Table 13: Fragility of mathematical reasoning under zero-shot CoT prompting on GSM8K vs. GSM-Symbolic benchmarks.

| Tasks | Qwen2.5-1.5B-Inst | Qwen2.5-Math-1.5B-Inst | Qwen2.5-Math-7B-Inst |
|---|---|---|---|
| GSM8K | 65.1% | 81.8% | 95.3% |
| GSM-Symbolic | 54.8% | 75.8% | 90.0% |
| GSM-P1 | 32.0% | 61.6% | 76.8% |
| GSM-P2 | 5.0% | 47.0% | 68.0% |

**The effect of in-context learning on LoRA routing.**   Table 14 shows the effect of in-context learning on LoRA routing methods evaluated on the GSM-P2 benchmark. The in-context examples are taken from the GSM8K, as the standard benchmarking procedures (Beeching et al., 2023; Yang et al., 2024a). In the baseline (non-LoRA) models, we observe that moving from 0-shot to 8-shot prompting yields little to no improvement, and in some cases, a slight degradation, in performance. For example, the Qwen2.5-Math-7B-Inst model drops from 0.68 (0-shot) to 0.42 (8-shot). This pattern aligns with prior findings reported in (Mirzadeh et al., 2024), which demonstrate that in-context learning with GSM8K exemples offers limited benefit on the GSM-P2 benchmarks. Applying LoRA routing methods such as Uniform and Arrow, generally reduces performance compared to the base models, especially for the larger models. When using 8-shot in-context examples, all models show decreased accuracy overall, with the base models again outperforming the merged variants. This suggests that LoRA routing, in the current setup, may not effectively preserve or enhance model performance in compositional generalization tasks, and that in-context learning does not compensate for these declines.

Table 14: Accuracy on GSM-P2 after routing LoRAs individually fine-tuned on GSM-Symbolic and GSM-P1, with and without in-context learning examples.

| Number of In-Context Examples | LoRA Routing Methods | Qwen2.5-1.5B-Inst | Qwen2.5-Math-1.5B-Inst | Qwen2.5-Math-7B-Inst |
|---|---|---|---|---|
| 0-shot | Base Model Only | 5% | 47% | 68% |
| | Uniform | 10% | 24% | 34% |
| | Arrow | 9% | 19% | 27% |
| 8-shot | Base Model Only | 0% | 26% | 42% |
| | Uniform | 8% | 19% | 33% |
| | Arrow | 5% | 18% | 6% |

**The Change of Base Model's Behavior After Routing LoRAs.**   To analyze the impact of LoRA merging or routing on model behavior, we examined the Qwen2.5-Math-1.5B-Inst model's zero-shot CoT outputs on the GSM-P2 evaluation set before and after routing. Using GPT-4o as an evaluator, we classified each solution as natural language, programming language, or unidentifiable. GPT-4o also assessed the correctness of each answer and, for incorrect outputs, categorized the error type as semantic misunderstanding, calculation error, or step-missing—following the definitions and prompting protocol of Zhong et al. (2025). Errors that did not fit any of these categories were labeled as unidentifiable, while correct answers were assigned the label *None.* To assess the reliability of GPT-4o's judgments, we measured its agreement with the exact-match metric on answer correctness.

Table 15 shows that routing LoRAs trained on natural language solutions can hinder the base model's ability to solve math problems with programming language (i.e., tool-integrated reasoning). Uniform and Arrow merge methods produced 100% natural language solutions, while the base model originally generated 12%

code-based responses. Both routing methods created high calculation errors (56%). These results suggest that while routing LoRAs trained on natural language solutions can impair domain-specialized math models' ability to solve problems using code. Thus, LoRA routing should consider the model's pretraining history and task-specific needs, as improper strategies could degrade its math problem-solving effectiveness.

Table 15: Analysis of the type of generated solutions and errors using GPT-4o as the judge.

| LoRA Routing Methods | Solution Types | | | Error Types | | | | | Agreement |
|---|---|---|---|---|---|---|---|---|---|
| | Natural Lang. | Code | Unident. | Semantic | Calc. Error | Step-Miss | Unident. | None (Correct) | |
| None | 87% | 12% | 1% | 16% | 29% | 6% | 1% | 48% | 97% |
| Uniform | 100% | 0% | 0% | 11% | 56% | 9% | 0% | 24% | 98% |
| Arrow | 100% | 0% | 0% | 17% | 56% | 12% | 0% | 15% | 96% |

## A.5 Detailed experimental setups and results for cross-lingual code generation task

**Setup** To evaluate whether our conclusions generalize beyond symbolic logical composition, we study a cross-lingual, cross-domain code generation task. Specifically, we assess whether separately trained adapters for English code generation and English $\rightarrow$ Spanish translation can be composed in a data-free manner, using Phi-2, Phi-3-mini-4k-instruct, Qwen2.5-7B, Qwen2.5-14B, the MBPP-Translated splits, and the Uniform and Arrow merging methods.

1. **Coding Adapter $L_1$ (English $\rightarrow$ Code Generation):** Trained on the English subset of the MBPP-Translated dataset AnonymousDoe, 2024.

2. **Translation Adapter $L_2$ (English $\rightarrow$ Spanish):**
   - **Phi 2 / Phi 3:** We reuse an EN–ES ParaCrawl-based adapter from the 256-task FLAN v2 library (Ostapenko et al., 2024), trained solely on general-domain parallel data.
   - **Qwen2.5 7B / 14B:** Fine-tuned on the same dataset as above using the hyperparameters from our main paper.

**Evaluation** We evaluate on the Spanish MBPP subset of the MBPP-Translated dataset using two data-free merging strategies: **Uniform** and **Arrow**.

**Results** As shown in Table 16, our findings from symbolic tasks are strongly supported by this cross-lingual code generation experiment:

1. **Phi models (English-only pretraining):** The translation adapter $L_2$ drives almost all improvement. Merging does not unlock new capabilities, reaffirming our earlier conclusion.

2. **Qwen2.5 models (multilingual pretraining):** Merging yields only marginal gains over single adapters.

Table 16: Compositional Performance on Cross-Lingual Code Generation (English Instructions $\rightarrow$ Spanish MBPP Code)

| Model | Base Only | Coding ($L_1$) | Translation ($L_2$) | Merged (Uniform) | Merged (Arrow) |
|---|---|---|---|---|---|
| Phi-2 | 0.0% | 1.8% | 36.0% | 28.1% | 28.8% |
| Phi-3-mini-4k-instruct | 0.0% | 2.3% | 37.2% | 27.9% | 30.5% |
| Qwen2.5-7B-Instruct | 32.9% | 34.0% | 37.5% | 34.9% | 35.5% |
| Qwen2.5-14B-Instruct | 38.7% | 38.9% | 44.0% | 39.1% | 40.2% |

The dominance of the translation adapter ($L_2$) and the degraded performance in merged configurations indicates a failure of composition. To ensure this failure is not due to a weak coding adapter, we also evaluate $L_1$ on its native English MBPP task. As shown in Table 17, $L_1$ is highly effective on its original domain, confirming that the poor Spanish MBPP performance arises from failed composition, not weak components.

Table 17: Validation of English Coding Adapter ($L_1$) on Native Task (English MBPP)

| Model | Base Only | Coding ($L_1$) |
|---|---|---|
| Phi-2 | 56.0% | 78.1% |
| Phi-3-mini-4k-instruct | 60.3% | 82.3% |
| Qwen2.5-7B-Instruct | 71.7% | 85.5% |
| Qwen2.5-14B-Instruct | 76.2% | 89.0% |

**Conclusion and Impact**

The base model-dependent behavior observed here powerfully reinforces the methodological choice of beginning with a controlled two-hop synthetic setup: the synthetic environment isolates knowledge sources and enables clean attribution of which component contributes which capability. This isolation is crucial because, in complex, knowledge-intensive cross-domain reuse tasks, it is inherently difficult to construct genuinely independent subtasks - the base model often possesses some capacity in both domains (e.g., translation and coding, or history and geography), and the individual adapters themselves may learn overlapping pieces of the composite skill. We believe that this controlled approach, validated by its ability to analyze complex real-world failures, is an essential first step toward understanding and enabling viable data-free reuse.

### A.6 Generalization to Knowledge-Intensive Tasks (History/Geography)

In this section, we provided evidence that the **pattern-matching** mechanism extends to knowledge-intensive cross-domain reuse.

**Setup** We directly reuse openly shared LoRAs to test whether pattern-matching in LoRA reuse extends beyond symbolic logical composition to knowledge-intensive tasks. We specifically provide new results on history and geography QA. These LoRAs, fine-tuned on Phi-2 as the base model, were originally trained on KILT/HotpotQA (subtask of Flan v2 Longpre et al. (2023) for varying levels of knowledge complexity Ostapenko et al. (2024). We probe them on HotpotQA variants with geography/history content in two settings:

1. Original questions, and

2. Continent-level probing (requiring answers at the continent level).

If pattern-matching were the underlying mechanism for reusing via uniform merging or arrow routing, we would expect failures under the continent-level probing condition.

**Setup and example fine-tuning/evaluation questions** As shown in Table 18, the selected LoRA tasks span a range of reasoning complexities, from simple single-hop questions to multi-hop "final exam" style queries. These LoRAs were evaluated on 97 history/geography question-answer pairs taken from original KILT/ HotpotQA validation set whose answers contain a city or a country in two settings: 1) Original version, and 2) Probe to answer at **continent level**, as illustrated in the Table 19:

**Results using Phi2 as the base model**

As shown in Table 20, accuracy drops sharply under continent-level probing, reinforcing our claim that LoRA reuse relies on pattern-matching rather than genuine logical composition.

**1. Reusing LoRAs yield unstable results.** We observe changes in accuracy on original HotpotQA, which: e.g., uniform merging ∼46.3% vs arrow routing ∼8.0% when aggregating the four Hotpot LoRA tasks (excluding "formulate").

Table 18: KILT/HotpotQA training examples for openly shared LoRAs.

| LoRAs Task name | Example question (from KILT / HotpotQA) | Example answer |
|---|---|---|
| **kilt_tasks_hotpotqa_straighforward_qa**: simple, single-hop HotpotQA questions (no complex multi-document bridging) | "Who wrote the play "Romeo and Juliet"?" | "William Shakespeare" |
| **kilt_tasks_hotpotqa_complex_question**: each requires two or more facts to answer | "Which U.S. state is home to the university where Tim Cook earned his MBA?" | "North Carolina" |
| **kilt_tasks_hotpotqa_combining_facts**: These require two clearly separate facts, joined by explicit reasoning. | "Which continent contains the home country of the band ABBA?" | "Europe" |
| **kilt_tasks_hotpotqa_final_exam**: These are multi-hop, difficult, "exam-style" reasoning questions that often combines 3+ facts. | "The author of 'The Trial' was born in the capital of which modern country?" | "Czech Republic" |
| **kilt_tasks_hotpotqa_formulate**: multi-hop or semi-multi-hop question formulation, rephrasing, or query interpretation | "Rewrite a question that seeks the birthplace of the painter who created Guernica." | "In which city was the artist who painted Guernica born?" |

Table 19: Examples of geography/history related HotpotQA questions and their corresponding continent-level probing prompts with answers.

| Original question (from KILT / HotpotQA) | Original answer | Continent Prompt question | answer |
|---|---|---|---|
| **"Where was the world cup hosted that Algeria qualified for the first time into the round of 16?"** | **"Brazil"** | **"On which continent is the location related to the following question: Where was the world cup hosted that Algeria qualified for the first time into the round of 16?"** | **"South America"** |
| **"In what city was the Italian Baroque composer who composed a set of sonatas, titled Op. 5, born?"** | **"Venice"** | **"On which continent is the location related to the following question: In what city was the Italian Baroque composer who composed a set of sonatas, titled Op. 5, born?"** | **"Europe"** |

**2. Pattern-matching as the underlying mechanism.** When prompted to answer the same question at the continent level, accuracy greatly dropped. This supports our claim that successfully reusing LoRAs on unseen tasks is driven by pattern-matching instead of genuin logical composition.

**Impact of our results**

This results highlight the risk of reusing openly shared LoRAs without data access since this approach was driven mainly by pattern-matching rather instead of geniue logical composition. By combining rigorous theoretical analysis with well-controlled experiments, we deliver the first mechanism-based account of when and why LoRA reuse can (or cannot) substitute for data access. This is a critical insight for practitioners

Table 20: Accuracy results for LoRA reuse on original HotpotQA questions versus continent-level probing using Phi-2 as the base model.

| Merge or routing methods | Phi2 / Original Question | | | Phi2 / Continent Prompt | | |
|---|---|---|---|---|---|---|
| | Base | Uniform | Arrow | Base | Uniform | Arrow |
| kilt_tasks_hotpotqa_straighforward_qa | 30.90% | 6.10% | 3.09% | 1.03% | 2.06% | 1.03% |
| all 4 hotpotqa (excl. formulate) | – | 46.30% | 8.00% | – | 5.15% | 2.06% |
| all 5 hotpot qa | – | 40.20% | 11.30% | – | 4.12% | 0.00% |

operating under data-sharing constraints. It's also important for academic researchers to explore other mechanisms with well-controlled experiments.

