# OpenReview forum: "When Does LoRA Reuse Work? Theoretical Limits and Mechanisms for Recycling LoRAs Without Data Access"
_TMLR — Accepted by TMLR_

### Review · Reviewer_oCHB · 2025-11-28

**Summary Of Contributions:**

This paper studies when and how LoRA adapters trained on different tasks can be reused or combined, without access to the original training data, to solve new tasks. On the theory side, the authors analyze a one-layer transformer with LoRA applied to the MLP layers and show that rank-one LoRA updates corresponding to different one-hop relations do not compose to yield correct two-hop reasoning. On the empirical side, the authors construct synthetic two-hop reasoning datasets and run experiments with Qwen2.5 and DeepSeek models ranging from 1.5B to 70B parameters. The authors observe that data-free merging or routing methods (Uniform, Arrow, TIES) rarely realize two-hop composition. However, performance improves when the fine-tuning data includes explicit bridge solution templates that link the one-hop tasks to the target composition.

**Audience:**

Yes

**Audience Explanation:**

* LoRA-based fine-tuning and reuse of adapters are widely used in practice and have received much attention in recent works on parameter-efficient fine-tuning (PEFT). This paper combines theoretical analysis and thorough experiments on when such reuse fails and under what conditions it starts to work, which will interest readers working on PEFT, domain adaptation, and model interpretability probably.

* The two-hop results provide a guideline for researchers designing adapter libraries: naive merging or routing is unlikely to produce promising multi-hop reasoning, even when each adapter solves its own task very well. The experiments on GSM benchmarks connect these ideas to math reasoning and highlight interactions between LoRA reuse again.

* The observations also help set expectations about what LoRAs can do without additional training data, thereby contributing to future research on domain adaptation and test-time learning.

**Broader Impact Concerns:**

This paper did not provide a broader impact statement. I did not see major ethical issues.

**Claims And Evidence:**

Yes

**Claims Explanation:**

* The core claims about data-free composition of LoRAs are supported by both theory analysis and corresponding experiments. Two-hop experiments across six base models and three dataset variants (F, H, R) consistently show near-perfect performance for single-task experts, but less than 10% accuracy on the composed A→C relation for merged or routed LoRA libraries, in line with theoretical predictions.

* The claims that bridge datasets with explicit templates can unlock reuse are also supported by experiments. More specifically, the authors vary whether CoT templates for the A→C relation appear in the bridge, whether these templates are shared between experts, and whether the bridge covers the same relations or alternative ones. The results obtained match the claims.

* However, the theoretical model is a bit simple (one layer and LoRA applied only to MLP layers). And all empirical tasks are synthetic, focused on two-hop reasoning, and based on GSM benchmarks. The evidence for generality is not sufficiently convincing.

**Requested Changes:**

* It would be helpful to extend the theoretical analysis to multi-layer settings to make it more convincing.

* In the current version, all empirical results are on synthetic two-hop reasoning with small entity sets and on GSM-based benchmarks. It would be better to add at least one additional task family or a non-math reasoning task, for example, coding generation.

* Most related works are discussed only conceptually, such as LoRA Lego, LoRI, Self MoE, and FLiX. It would strengthen the contribution to run at least one or two of these on a representative two-hop setting.

* About clarity. Some important details are only in the appendices. For instance, a summary of training hyperparameters for two-hop tasks is suggested to be included in the main text (*e.g.*, a short table summarizing learning rates, batch sizes, optimizer, etc.).

---

> ### Author Response · Authors · 2025-12-13
> **Replies to recommended change 1.**
>
> We appreciated the reviewer's recognition of the importance of our contribution to future research and practitioners. We address the requested changes:
>
> ### 1. Theoretical analysis
>
> > •	It would be helpful to extend the theoretical analysis to multi-layer settings to make it more convincing.
>
> We agree that such an extension would be desirable; however, current mathematical techniques for theoretical understanding of transformers do not make this feasible.
> Our analysis is based on Nichani et al (2025) (cited in the paper), a theoretical study of factual recall in transformers, which makes similar simplifications (such as a one-layer setup) as our study.
> We now discuss this in detail in Appendix A.1.2.
> We furthermore would like to emphasize that the experiments in Section 4 demonstrate the relevance of our conclusions to real-world, multi-layer LLMs.

---

> > ### Author Response · Authors · 2025-12-13
> > **Replies to recommended change 2.**
> >
> > ### 2. Our conclusion extends to cross-lingual code generation task
> >
> > > •	In the current version, all empirical results are on synthetic two-hop reasoning with small entity sets and on GSM-based benchmarks. It would be better to add at least one additional task family or a non-math reasoning task, for example, coding generation.
> >
> >
> > Thank you for this valuable suggestion. Using the synthetic 2-hop and GSM datasets, we discovered that 1) merging/routing performance heavily depends on base-model’s pretraining history, and 2) reusing LoRAs barely unlocks new capabilities. To examine whether these conclusions extend beyond symbolic logical composition, we added a cross lingual, cross domain code generation experiment.
> >
> > **Setup:**
> >     1. Coding Adapter $L_1$ (English -> Coding): trained on the “English” subset of MBPP Translated dataset (AnonymousDoe/MBPP Translated on Hugging Face).
> >     1. Translation Adapter $L_2$ (English -> Spanish):
> > •	Phi 2 / Phi 3: We reuse an EN–ES ParaCrawl based adapter from the 256 task Flan v2 library (Ostapenko et al., 2024), trained solely on general domain parallel text.
> > •	Qwen2.5 7B and Qwen2.5 14B: fine-tuned on the above dataset using the same hyper-parameters as in our paper.
> >
> > **Evaluation** We test on the Spanish MBPP subset of the above mentioned MBPP-Translated dataset by reuse them with two data-free methods, Uniform and Arrow.
> >
> > **Results**
> >
> > Our claims supported by the **cross‑lingual code generation task**:
> > 1. For Phi models (English only pretraining), the translation LoRA drives most of the improvement, supporting our claim that merging LoRAs does not unlock new capabilities.
> > 1. For Qwen2.5 (multilingual pretraining), merging yields only marginal gains over single adapters.
> >
> > **Table.** Compositional Performance on Cross-Lingual Code Generation (English Instructions to Spanish MBPP Code)
> >
> > | Model                     | Base only | Coding Adapter ($L_1$) | Translation Adapter ($L_2$) | Merged (Uniform) | Merged (Arrow) |
> > |---------------------------|-----------|----------------|---------------------|-------------------|----------------|
> > | Phi-2                     | 0.0%      | 1.8%           | 36.0%               | 28.1%             | 28.8%          |
> > | Phi-3-mini-4k-instruct    | 0.0%      | 2.3%           | 37.2%               | 27.9%             | 30.5%          |
> > | Qwen2.5-7B-Instruct       | 32.9%     | 34.0%          | 37.5%               | 34.9%             | 35.5%          |
> > | Qwen2.5-14B-Instruct      | 38.7%     | 38.9%          | 44.0%               | 39.1%             | 40.2%          |
> >
> > The clear performance dominance of the Translation Adapter ($L_2$) and the performance degradation in merged configurations suggests a failure in composition. To rigorously validate that this failure is due to the merging mechanism and not an intrinsically weak component, we evaluated the Coding Adapter ($L_1$) on its native task: English code generation (English MBPP). The results below confirm that $L_1$ is highly effective in its specialized domain. This establishes that the poor performance on Spanish MBPP when combined with the Coding Adapter is not due to a weak adapter, but rather the inability to compose it effectively with the Translation Adapter ($L_2$).
> >
> > **Table.** Validation of English Coding Adapter ($L_1$) on Native Task (English MBPP)
> >
> > | Model                     | Base only | Coding Adapter ($L_1$) |
> > |---------------------------|-------------------------|---------------|
> > | Phi-2                     | 56.0%                   | 78.1%         |
> > | Phi-3-mini-4k-instruct    | 60.3%                   | 82.3%         |
> > | Qwen2.5-7B-Instruct       | 71.7%                   | 85.5%         |
> > | Qwen2.5-14B-Instruct      | 76.2%                   | 89.0%
> >
> >
> > **Conclusion and impact of our work**
> > The base model-dependent behavior observed here powerfully reinforces the methodological choice of beginning with a controlled two-hop synthetic setup: the synthetic environment isolates knowledge sources and enables clean attribution of which component contributes which capability. This isolation is crucial because, in complex, knowledge-intensive cross-domain reuse tasks, it is inherently difficult to construct genuinely independent subtasks - the base model often possesses some capacity in both domains (e.g., translation and coding, or history and geography), and the individual adapters themselves may learn overlapping pieces of the composite skill. We believe that this controlled approach, validated by its ability to analyze complex real-world failures, is an essential first step toward understanding and enabling viable data-free reuse.

---

> > > ### Author Response · Authors · 2025-12-13
> > > **Replies to recommended change 3.**
> > >
> > > ### 3. Latest data-free merge method failed on 2-hop
> > >
> > >
> > >
> > > > •	Most related works are discussed only conceptually, such as LoRA Lego, LoRI, Self MoE, and FLiX. It would strengthen the contribution to run at least one or two of these on a representative two-hop setting.
> > >
> > >
> > > Due to limited time, we extended the 2-hop experiment shown in Table 2 to “LoRI”, which is the only one among the suggested list that both has openly shared code and is specifically designed to reduce interference when combining multiple adapters without access to training data.
> > > The two variants of LoRI (LoRI-D/S) can be merged in two ways:
> > > 1.	Linear merging: a single adapter is obtained by taking a fixed linear combination of the task adapters. Our Uniform baseline is simply the special case where all tasks receive the same fixed weight.
> > > 2.	Concatenated merging: the task-specific adapters are instead stacked into a larger block-structured adapter, so that each task mainly occupies its own subspace.
> > >
> > > **Result as an extension of our Table 2**
> > >
> > > **Table.** Performance of LoRA/LoRI-based routing/merging methods using a 2-combination adapter library on Qwen2.5-7B with the R dataset.
> > >
> > > | Adaptation | Routing/Merging | Accuracy |
> > > |-----------|-----------------|----------|
> > > | LoRA      | Uniform         | 3%       |
> > > | LoRA      | Arrow           | 3.9%     |
> > > | LoRA      | TIES            | 3.8%     |
> > > | LoRA      | CAT (LoRA Soup)  | **21%**  |
> > > | LoRI-D    | Concat          | 13%     |
> > > | LoRI-D    | Linear          | 1%       |
> > > | LoRI-S    | Concat          | 7%     |
> > > | LoRI-S    | Linear          | 3%     |
> > >
> > > **These results support our main claim:** simply merging single hop adapters without CoT-bridge or dedicated two hop training is insufficient for reliable two hop generalization, even with more sophisticated architectures like LoRI. All LoRI variants achieve low accuracy. The best one (LORI-D Concat) still underperforms CAT, which required access to original fine-tuning data to determine the routing.

---

> > > > ### Author Response · Authors · 2025-12-13
> > > > **Replies to recommended change 4.**
> > > >
> > > > ### 4. Writing style
> > > >
> > > >
> > > > > •	About clarity. Some important details are only in the appendices. For instance, a summary of training hyperparameters for two-hop tasks is suggested to be included in the main text (e.g., a short table summarizing learning rates, batch sizes, optimizer, etc.).
> > > >
> > > >
> > > > Thank you for pointing this out. Given the short timeline of the discussion period and to keep the main paper focused, we are currently keeping the full details in the appendix in the updated PDF. We can add them later if it still would be necessary.

---

> > > ### Comment · Reviewer_oCHB · 2025-12-13
> > >
> > > Thanks the authors for the rebuttal and the revisions. The changes, especially the new results on code generation, have strengthened the manuscript further. My concerns are addressed, and I am happy to recommend acceptance.

---

### Review · Reviewer_SUrz · 2025-11-29

**Summary Of Contributions:**

This paper explores the limits of LoRA reuse without data access. Through experiments on reasoning/math tasks, it suggests that standard merging methods (like Uniform or Arrow) generally fail for compositional tasks. "Success" in their setups is largely observed when "bridge" templates (shared CoT structures) are present in the training data.

**Audience:**

Yes

**Audience Explanation:**

LoRA reuse and merging are widely used in practice, and this paper offers a clear takeaway on when composition fails and what kinds of template alignment make it work.

**Broader Impact Concerns:**

No concerns

**Claims And Evidence:**

Yes

**Claims Explanation:**

The theoretical derivation gives a solid intuition for the subspace mismatch. The experimental controls are rigorous. And the paper is generally honest about the limitations (e.g., acknowledging performance degradation in specialized models), which adds credibility.

**Requested Changes:**

I recommend acceptance. This paper serves as a valuable "reality check" for the field. I have a few minor suggestions that I think would strengthen the final manuscript.

1. Table 14 offers a very interesting insight. The Base Model utilizes code-based responses in roughly 12% of cases, but the Merged models (Uniform/Arrow) appear to drop this to 0%, reverting entirely to Natural Language. This suggests the merging process might be suppressing the specialized "tool use" capability, biasing the model back to the dominant modality (text) of the individual LoRAs. Explicitly discussing this mechanism in the error analysis would add significant depth.

2. The positive results in Section 4.1.2 rely on the LoRAs being trained with the specific CoT structure of the target task. It might be beneficial to restate in the Discussion that this acts as a form of format/template exposure. Since effective reuse seems to require injecting the target reasoning pattern into the LoRA training, clarifying this constraint would help practitioners better understand the practical limits of "data-free" reuse.

---

> ### Author Response · Authors · 2025-12-13
> **Reiterating Key Findings and Practical Limits of Data‑Free Reuse**
>
> We thank the reviewer for valuing our work as a valuable reality check. We appreciate the reviewers’ suggestion. To help practitioners to understand the current limits and design systems to enable “data-free” reuse in the future, we address the two suggestions in our revision as follows (blue texts in newly uploaded PDF):
>
> 1.	Re-iterate the findings from referred Table in the “Discussion” section. In the “Conclusion” section, we emphasize the importance of alignment of pre-training and fine-tuning dataset as a pre-condition for practitioners to prepare or design system aiming for utilizing special capability of the base model.
>
> 2.	Re-state the positive results from Section 4.1.2 in the “Discussion” section to help practitioners better understand the practical limits of “data-free” reuse.

---

### Review · Reviewer_arGZ · 2025-11-29

**Summary Of Contributions:**

This paper studies whether parameter-efficient fine-tuning modules (LoRAs) can be reused for new tasks without access to original training data. Through a combination of theoretical analysis and empirical evaluations across two representative logical reasoning tasks (two-hop symbolic reasoning and math word problems), the authors show that simply merging or routing LoRAs generally fails to achieve true compositional generalization. The paper further identifies that reuse only succeeds when LoRA training data share reusable solution patterns (e.g., shared CoT or executable code), suggesting that LoRA reuse primarily exploits surface-level template matching rather than genuine reasoning combination. These findings provide a valuable mechanistic understanding of the current limitations of LoRA-based model reuse.

Strengths:

1.Addresses an important and timely problem regarding LoRA reuse under data-restricted settings.

2.Clear theoretical insights supported by controlled experiments across different model scales.

3.Provides actionable takeaways for the community (e.g., the importance of reusable reasoning templates).

Weaknesses:

1.Task scope is limited to symbolic logical composition. It remains unclear whether the conclusions generalize to more semantic or knowledge-intensive scenarios (e.g., combining independently trained history and geography LoRAs for cross-domain QA).

2.Success depends strongly on bridge data. When CoT/code templates are introduced, the setting shifts from “pure LoRA reuse without data” to providing additional compositional supervision. Clearer clarification of this boundary would improve the framing.

3.Theory–practice gap not fully addressed. The theoretical analysis is based on a single-layer abstraction; empirical discussion on how multi-layer LLM structures might interact with LoRA interference would strengthen generality of the conclusions.

4.Evaluation focuses mainly on final answer accuracy. Additional analysis of reasoning faithfulness or step-level correctness would better support the claim about failure to achieve true compositionality.

**Audience:**

Yes

**Audience Explanation:**

The topic is highly relevant to the TMLR community. While LLMs are increasingly powerful, their training and adaptation costs remain substantial. Understanding whether strong performance on new tasks can be achieved by reusing existing LoRA modules—without accessing original training data or performing expensive retraining—is both practically meaningful and scientifically important. The insights provided here about when LoRA reuse succeeds or fails will directly benefit researchers and practitioners who rely on PEFT methods for scalable model deployment.

**Broader Impact Concerns:**

No major ethical concerns. This work focuses on understanding the limitations of LoRA reuse and does not introduce new deployment risks.

**Claims And Evidence:**

Yes

**Claims Explanation:**

The paper’s main claims are well supported through a combination of theoretical analysis and carefully controlled experiments. The synthetic two-hop reasoning task and the symbolic math setting are both designed to isolate compositional ability, and the results consistently validate the theoretical prediction that simple merging or routing of LoRAs fails to produce true compositional generalization. The use of different model scales, different entity familiarity conditions, and bridge-data ablations further strengthens the empirical rigor. Moreover, the behaviors observed across multiple settings all align with the proposed mechanism that LoRA reuse predominantly captures surface-level template matching rather than logical combination of knowledge.

**Requested Changes:**

The first two weaknesses are critical: (1) clarifying the scope of the conclusions beyond symbolic logical composition (e.g., whether the findings extend to knowledge-intensive cross-domain reuse such as history+geography QA), and (2) more clearly defining the role and boundary of bridge data, since its introduction adds compositional supervision beyond pure LoRA reuse. Additionally, discussing alignment between the single-layer theory and multi-layer LLM behavior, and including some form of reasoning-faithfulness analysis, would further strengthen the work but are not required for acceptance.

---

> ### Author Response · Authors · 2025-12-13
> **Regarding requested change - clarify scope**
>
> Thank you for pointing these out. While we updated the theoretical aspects in the uploaded pdf in blue, we would like to address the major weaknesses:
>
> ### 1. Clarify Our Scope
> Our study provides the first mechanism-based reality check on LoRA reuse without requiring access to original training data. To balance the future research direction and current practical implication of our theoretical analysis, we used well-controlled experiment to show when failure and success can be observed, which can raise practitioners' awareness and motivate future research.
>
> We focus on compositional factual and reasoning tasks, rather than knowledge-intensive domains, precisely because this allows us to disentangle knowledge in the base model from the effect of composing different LoRAs to unlock composition, because we can ensure that correct answers can only obtained by composing knowledge from both LoRAs, and not by retrieving knowledge already present in the base model. We have made this more explicit in the abstract and the introduction, and will further revise to make it more salient in the paper.

---

> ### Author Response · Authors · 2025-12-13
> **Regarding requested change - findings can be extended to knowledge-intensive cross-domain reuse**
>
> ### 2. Findings can be extended to knowledge-intensive cross-domain reuse
>
> While our focus is on compositional factual and reasoning tasks, we now also report evidence that our findings extend to knowledge-intensive cross-domain reuse. We conduct an additional experiment directly reusing openly shared LoRAs to test whether pattern‑matching in LoRA reuse extends beyond symbolic logical composition to knowledge‑intensive tasks. We specifically provide new results on history and geography QA. These LoRAs, fine‑tuned on Phi‑2 as the base model, were originally trained on KILT/HotpotQA (subtask of Flan v2 (Longpre et al.)), 2023) for varying levels of knowledge complexity (Ostapenko et al., 2024). We test them on HotpotQA variants with geography/history content in two settings:
>
> 1. Original questions, and
> 2. Continent‑level probing (requiring answers at the continent level).
>
> If pattern‑matching were the underlying mechanism for reuse via uniform merging or arrow routing, we would expect failures under the continent‑level probing condition.
>
> **Setup and example fine-tuning/evaluation questions** The selected shared LoRAs were trained on KILT / HotpotQA tasks span single‑hop, multi‑hop, and “final exam” style reasoning, with examples requiring multiple facts across domains. Table below shows the selected LoRAs and their example training data, respectively:
> | LoRAs Task name                                 | Example question (from KILT / HotpotQA)                                                                                                                     | Example answer                |
> | ----------------------------------------- | ----------------------------------------------------------------------------------------------------------------------------------------------------------- | ----------------------------- |
> | **kilt_tasks_hotpotqa_straighforward_qa**: simple, single-hop HotpotQA questions (no complex multi-document bridging) | **“Who wrote the play “Romeo and Juliet”?”**                                                                                  | **“William Shakespeare”**       |
> | **kilt_tasks_hotpotqa_complex_question**: each requires two or more facts to answer  | **“Which U.S. state is home to the university where Tim Cook earned his MBA?”**               | **“North Carolina”**              |
> | **kilt_tasks_hotpotqa_combining_facts**: These require two clearly separate facts, joined by explicit reasoning.    | **“Which continent contains the home country of the band ABBA?”** | **“Europe”**                   |
> | **kilt_tasks_hotpotqa_final_exam**: These are multi-hop, difficult, “exam-style” reasoning questions that often combines 3+ facts.        | **“The author of 'The Trial' was born in the capital of which modern country?”**                         | **“Czech Republic”** |
> | **kilt_tasks_hotpotqa_formulate**: multi-hop or semi-multi-hop question formulation, rephrasing, or query interpretation         | **“Rewrite a question that seeks the birthplace of the painter who created Guernica.”**                  | **“In which city was the artist who painted Guernica born?”**           |
>
> **Evaluation** Those LoRAs were evaluated on 97 history/geography question-answer pairs taken from original KILT/ HotpotQA validation set whose answers contain a city or a country in two settings: 1) Original version, and 2) Probe to answer at **continent level**, as illustrated in the following table:
>
> | Original question (from KILT / HotpotQA) | Original answer   |  Continent Prompt question  |  answer   |
>  --------------------------| ----------------------------- | ------------------ | ------------------ |
> | **“Where was the world cup hosted that Algeria qualified for the first time into the round of 16?”** | **“Brazil”**       | **“On which continent is the location related to the following question: Where was the world cup hosted that Algeria qualified for the first time into the round of 16?”** | **“South America”** |
> | **“In what city was the Italian Baroque composer who composed a set of sonatas, titled Op. 5, born?”** | **“Venice”**       | **“On which continent is the location related to the following question: In what city was the Italian Baroque composer who composed a set of sonatas, titled Op. 5, born?”** | **“Europe”** |

---

> ### Author Response · Authors · 2025-12-13
> **Results - findings can be extended to knowledge-intensive cross-domain reuse**
>
> **1. Reusing LoRAs yield unstable results.** We observe changes in accuracy on original HotpotQA, which: e.g., uniform merging ~46.3% vs arrow routing ~8.0% when aggregating the four Hotpot LoRA tasks (excluding “formulate”).
>
> **2. Pattern-matching as the underlying mechanism.** When prompted to answer the same question at the continent level, accuracy greatly dropped. This supports our claim that successfully reusing LoRAs on unseen tasks is driven by pattern-matching instead of genuin logical composition.
>
>
> | merge or routing methods                         | **Phi2/Original Question** |               |               | **Phi2/Continent Prompt** |               |               |
> |--------------------------------------------------|----------------------------:|--------------:|--------------:|--------------------------:|--------------:|--------------:|
> |                                                  | base                        | uniform       | arrow         | base                      | uniform       | arrow         |
> | kilt_tasks_hotpotqa_straighforward_qa            | 30.90%                      | 6.10%         | 3.09%         | 1.03%                     | 2.06%         | 1.03%         |
> | all 4 hotpotqa (excl. formulate)                 |                              | 46.30%        | 8.00%         |                           | 5.15%         | 2.06%         |
> | all 5 hotpot qa                                  |                              | 40.20%        | 11.30%        |                           | 4.12%         | 0.00%         |
>
> We furthermore conducted a second set of experiments to further demonstrate the scope of our results, in evaluating combination of LoRAs representing translation and coding abilities; we refer to our response to **Reviewer oCHB** for the details.
>
>
> ### Impact of our paper and Role of Bridge Method
> Our results highlight challenges in reusing openly shared LoRAs without data access, since resulting performance appears to be driven mainly by pattern-matching instead of genuine logical composition. By combining theoretical analysis with well-controlled experiments, we deliver the first mechanism-based account of when and why LoRA reuse can (or cannot) substitute for data access. This is a critical insight for practitioners operating under data-sharing constraints. It's also important for academic researchers to explore other mechanisms with well-controlled experiments.
> In this context, the role of the **bridge method** is to demonstrate a successful situation that enables composition that otherwise fails when directly combining LoRAs. This finding can offer actionable guidance for practitioners preparing fine-tuning datasets, illustrating how to design reasoning templates as bridges for composing abilities from different LoRAs.  We have expanded the Discussion to make this explicit.
>
> ----
> Ostapenko, O., Su, Z., et al. (2024) Proceedings of the 41st International Conference on Machine Learning, PMLR 235:38885-38904, .
>
> Longpre, S., Hou, L., Vu, T., Webson, A., Chung, H. W., Tay, Y., Zhou, D., Le, Q. V., Zoph, B., Wei, J., & Roberts, A. (2023). The Flan Collection: Designing Data and Methods for Effective Instruction Tuning.

---

### Review · Reviewer_54XL · 2025-12-05

**Summary Of Contributions:**

**Summary:**

LoRA recycling/reuse refers to the idea of combing LoRA adapters from independent fine-tuning jobs to solve new tasks by combining the knowledge from the component adapters. Many methods exist for reuse, but there is little work on understanding the mechanism behind these methods or why & when they work. This work studies when LoRA reuse without access to training data can be useful using a simplified theoretical model and a series of empirical experiments on two-hop reasoning and math reasoning.

On the theory front, the authors consider a highly simplified transformer model consisting of a *frozen* randomly initialized embeddings and single attention layer followed by a trainable classification MLP. They consider the setting of combining two LoRAs trained on two separate relations (A->B => B is spouse of A; B->C => B lives in C ) to perform two-hop reasoning (A->C) that requires knowledge from both LoRAs. They demonstrate that averaging adapters or methods like CAT or Arrow cannot result in compositional generalization in their specific setup.

On the empirical side, they conduct experiments on publicly available LLMs with two-hop reasoning and math reasoning. The two-hop reasoning is similar to the setup in the theoretical analysis where they investigate whether merging LoRAs for A->B and B->C can lead to inference of A->C. Their results show that merging the two LoRA adapters leads to very poor performance on the A->C task. Additionally, they also find that using familiar entities/relations from the pretraining data helps improve performance, and methods like CoT are ineffective in addressing the poor performance of merged adapters. They arrive at similar conclusions for the math reasoning experiments.

Their main findings overall are: (i) LoRA reuse is ineffective regardless of model and data size, (ii) it's not possible to reliably merge LoRAs from logically disjoint datasets, and you need overlap like similar reasoning patterns for LoRA reuse to be effective, (iii) LoRAs depend more on shallow pattern matching than compositional generalization.

**Weaknesses:**
1. The paper is easy to understand up to section 3. Section 4 has a very high cognitive load and I found it difficult to follow. There are so many variables and dimensions to keep track of. Many important experimental details are in the Appendix A2 instead of the maintext. I had to reread several times and go back and forth to the appendix to understand the section. See later comments for specific instances.
2. The theoretical model isn't a convincing model for or hasn't been motivated enough. The assumptions appear to be gross simplifications of an actual transformer; e.g., only training the final classification weights, uniform softmax logits, etc. I'm also not sure whether the random features model being a transformer has any effect on the analysis. Could it have been any random neural network that can mix across tokens?
3. The theoretical model focus primarily on retrieving/editing factual information as the LoRA parameters are likely learning to add/edit factual information in the weights. In such scenarios, it naturally feels like LoRA compositions like deducing transitive relationships would be unlikely to work. Is this the only mechanistic view of LoRA parameters?

**Additional Comments:**

**Questions:**
1. What are “noise tokens” in section 3 introduction?
2. Were the number of LoRA parameters kept constant for Figure 1 models? Is it possible that the LoRA parameters are increasing with model size and the models are better able to memorize the training data in combination 3 setting? What stops the LoRA parameters from memorizing all A -> C pairs?
3. In A.1, Eq (5), why is the current token not allowed to attend to itself? Casual language models often allow the tokens to attend to all past and itself. It also appears like the analysis relies on this fact.
4. Would implications of LoRA on W in section 3 mean anything for LoRA on attention/MLP blocks in transformers? Transformers may rely on a completely different mechanism than LoRA on singled out linear layers. For example, the LoRA parameters of the two feedforward layers in MLP blocks of transformers could be coupled and work in tandem.
5. Why is the proposed theoretical model a good proxy for a real transformer? It appears to be an extremely simplified version of a transformer. While analyzing an actual transformer might be practically intractable, I think there should be convincing explanation for why the results from these simplified models transfer to real models.
6. “We follow this assumption, and additionally take U, V to remain untrained, as we do not assume noise tokens in the context.” What does this mean?

**Audience:**

Yes

**Audience Explanation:**

I'm not convinced about the value of the theoretical work here but there certainly are people who be interested in it. There are many empirical findings that are interesting such as familiarity playing a substantial role in the ability to reuse adapters.

**Broader Impact Concerns:**

N/A.

**Claims And Evidence:**

No

**Claims Explanation:**

Somewhat. The claims in the abstract and introduction are very broad and general, whereas the experiments and the theoretical analysis study very specific settings. I am not convinced that their conclusions generalize to all usecases of LoRA reuse/recycling; e.g., altering personalization of an LLM by varying mixtures of LoRAs likely does not use the factual retrieval/editing mechanism the paper is concerned with.

**Requested Changes:**

**Minor writing related comments:**
1. The main text does not mention how the 2-combination and 3-combinations adapters were obtained in 4.1. The appendix suggests Arrow and uniform averaging were tested and the authors stuck with Arrow, but I think the method should be made clear in the maintext.
2. A concrete example of the inputs and targets from A.2.1 would be helpful in section 4.1. I had to reread a few times and then visit A.2.1 to understand what was happening; for a moment, I thought <name, location> were the items in the dataset and was completely lost.
3. It might be useful to mention there is a residual connection and casual modeling w/o attending to self in Section 3. Proposition 1 looks a bit sus on first glance while attempting to eyeball the expressions to the forward pass expressions. It's not clear without looking at the appendix, but it would be easier to interpret had the residual connection and causal masking w/o self was mentioned.
4. What about trainability of K, Q in section 3? The appendix mentions that K, Q are randomly init and left untrained, but is omitted in the maintext for some reason while others weight matrices are explicitly discussed.

**A non-exhaustive list of nitpicks:**
1. "To address this, we began with the theoretical analysis *revealed* its limit."
2. "LoRA reuse can be successful without direct data-access when the common solution templates (e.g.,
chain-of-thoughts reasoning patterns or re-usable code snippets) *existed* in the fine-tuning datasets
as bridges."
3. A.2.2: "This is a special case of LoraHub (Huang et al., 2024) *axssigning* each LoRA adapter the same weight."
4. 4.1.1: "in smaller models and the presence *of of* unfamiliar entities"

---

> ### Author Response · Authors · 2025-12-13
> **Regarding weaknesses**
>
> Thank you for the close reading and your feedback, which we really appreciate. We have closely taken it into consideration in revising our paper, as detailed below.
>
> >The paper is easy to understand up to section 3. Section 4 has a very high cognitive load and I found it difficult to follow. There are so many variables and dimensions to keep track of. Many important experimental details are in the Appendix A2 instead of the maintext. I had to reread several times and go back and forth to the appendix to understand the section. See later comments for specific instances.
>
> Thank you for this feedback. We will do our best to reduce cognitive load in Section 4, by (i) unpacking the different variables and dimensions in the writing, and (ii) bringing key experimental details into the main paper. Due to the tight timeline of the discussion period, we have not been able to do this yet, but will assign this a high priority in our further revision.
>
> > The theoretical model isn't a convincing model for or hasn't been motivated enough. The assumptions appear to be gross simplifications of an actual transformer; e.g., only training the final classification weights, uniform softmax logits, etc. I'm also not sure whether the random features model being a transformer has any effect on the analysis. Could it have been any random neural network that can mix across tokens?
>
> Indeed, we fully acknowledge that the theoretical model is highly simplified, in order to make theoretical analysis tractable at all given current methods. The motivation for the setup is based on in Nichani et al (2025) [1, cited in the paper], which provides a reference point for a model of factual retrieval in transformers, on which we build to integrate LoRA and LoRA merging.
>
> To address the reviewer's concern, we have revised the paper to clearly explain the motivation, the simplifying assumptions made, and be transparent about the fact that the model is simplified.
> Specifically, we have added Appendix A.1.2, where we provide an extensive discussion of how our setup is based on and compares to that of Nichani et al (2025). At a high level: both Nichani et al (2025) and our analysis simplify the model so it has only one nonlinearity left; this seems necessary to make theoretical understanding feasible. Our specific choices are guided by the need to keep the nonlinear MLP, which is needed to encode general functional relationships.
>
> We have added a pointer to this in the main paper, and have rephrased (including in the abstract) to make transparent that our theoretical analysis applies to a simplified setup.
>
> Regarding the last question (are the results specific to transformers?), our results apply more generally than to transformers, but not to fully arbitrary neural networks. We discuss this at the end of Appendix A.1.2.
>
> ---
> [1] Eshaan Nichani, Jason D. Lee, and Alberto Bietti. Understanding factual recall in transformers via associative memories. In The Thirteenth International Conference on Learning Representations, 2025.

---

> > ### Author Response · Authors · 2025-12-13
> > **Regarding weaknesses**
> >
> > >The theoretical model focus primarily on retrieving/editing factual information as the LoRA parameters are likely learning to add/edit factual information in the weights. In such scenarios, it naturally feels like LoRA compositions like deducing transitive relationships would be unlikely to work. Is this the only mechanistic view of LoRA parameters?
> >
> > Indeed, as the reviewer says, our theoretical analysis focuses on the editing of factual knowledge. We do this because this is a particular clean and well-formalizable model of what it might mean for a model to learn new composable skills. Our mechanistic view of LoRA parameters is especially similar to RoME (Meng et al, 2022) [2], as we acknowledge under Proposition 1. We acknowledge that updating factual knowledge is not the only function that LoRA adapters can perform. We discuss the relation of our analysis to mechanistic views on LoRA in Appendix A.1.3.
> >
> >
> > > Somewhat. The claims in the abstract and introduction are very broad and general, whereas the experiments and the theoretical analysis study very specific settings. I am not convinced that their conclusions generalize to all usecases of LoRA reuse/recycling; e.g., altering personalization of an LLM by varying mixtures of LoRAs likely does not use the factual retrieval/editing mechanism the paper is concerned with.
> >
> >
> > We agree that there are many use cases for re-using LoRAs, and personalization can be an interesting use case.
> >
> > Our work  deliberately focuses on **compositional factual and reasoning tasks**, where performance can be objectively measured through answer correctness.  This focus, especially the 2-hop experiments, allows us to **isolate knowledge from the base model** and obtain clean attribution of which component is responsible for which capability. It is actually quite hard to construct a genuinely two-step task where each LoRA is specialized in a subtask that is truly independent and unseen by both the base model and the other LoRA. In knowledge-intensive real-world settings, and even more so in “personalization” or “style” settings, this clean separation becomes almost impossible: the base model already exhibits non-trivial abilities in both domains (e.g., some translation, some coding, many stylistic registers), and style/persona LoRAs typically act by shaping global output behavior rather than encoding clearly localized factual relations. As a result, both the base model and the different LoRAs can share and overlap in what they learn, making it very difficult to attribute success or failure of a composed system to specific components. In contrast, our experiments (especially the 2-hop experiments), by introducing new knowledge, ensure that correct answers can only be obtained by composing abilities introduced by the adapters, not by activating abilities from the base model.
> >
> > We therefore clarify that our conclusions are **explicitly scoped** to symbolic composition (two-hop relations, math reasoning). We do not claim that our negative results automatically extend to all forms of LoRA reuse, especially not to stylistic or personalization mixtures, which likely operate via different underlying mechanisms (e.g., interpolation in a style space rather than computing a new symbolic mapping). We view a deeper mechanistic analysis of personalization-style mixtures as valuable and complementary future work. We have already rephrased the abstract and the introduction, and will further revise the paper to make the scoping more salient in the paper.

---

> > > ### Author Response · Authors · 2025-12-13
> > > **Regarding requested changes**
> > >
> > > > The main text does not mention how the 2-combination and 3-combinations adapters were obtained in 4.1. The appendix suggests Arrow and uniform averaging were tested and the authors stuck with Arrow, but I think the method should be made clear in the maintext.
> > >
> > > Thank you for this concrete feedback. We will make this change.
> > >
> > >
> > > > A concrete example of the inputs and targets from A.2.1 would be helpful in section 4.1. I had to reread a few times and then visit A.2.1 to understand what was happening; for a moment, I thought <name, location> were the items in the dataset and was completely lost.
> > >
> > > Likewise, we will make this change.
> > >
> > > > It might be useful to mention there is a residual connection and casual modeling w/o attending to self in Section 3. Proposition 1 looks a bit sus on first glance while attempting to eyeball the expressions to the forward pass expressions. It's not clear without looking at the appendix, but it would be easier to interpret had the residual connection and causal masking w/o self was mentioned.
> > >
> > > We now mention this explicitly (i) in Section 3.1, (ii) after Proposition 1.
> > >
> > > > What about trainability of K, Q in section 3? The appendix mentions that K, Q are randomly init and left untrained, but is omitted in the maintext for some reason while others weight matrices are explicitly discussed.
> > >
> > > Thanks for catching this. We added explicit discussion of K, Q together with the other matrices.

---

> > > > ### Author Response · Authors · 2025-12-13
> > > > **Answers to the additional questions**
> > > >
> > > > > What are “noise tokens” in section 3 introduction?
> > > >
> > > > This refers to a detail in the Nichani et al (2025) model: that model assumed that the prompt contains not just an entity and a relation, but also unrelated ``noise'' tokens that the model learns to ignore. We do not assume such tokens in our model. We have removed the reference from Section 3, and instead discuss the relation to the Nichani et al (2025) model in detail in Appendix A.1.2.
> > > >
> > > > > Were the number of LoRA parameters kept constant for Figure 1 models? Is it possible that the LoRA parameters are increasing with model size and the models are better able to memorize the training data in combination 3 setting? What stops the LoRA parameters from memorizing all A -> C pairs?
> > > >
> > > > The LoRA rank is kept fixed across model sizes, but the absolute number of LoRA parameters is not constant: it scales with the hidden size of the underlying model (since each adapter is a low-rank update of a wider weight matrix). However, our conclusions do not rely on limiting adapter capacity. We use the “combination 3” setting precisely as an upper bound: LoRA 3 is directly trained on A→C, and we then test whether a routing mechanism will correctly route to that adapter when the question is rephrased but refers to the same underlying fact. Indeed, we expect (intentionally) that the LoRA will memorize all pairs in this setup. In that case, performance reflects how well the router can exploit an adapter that already encodes the exact mapping, not a limitation of LoRA composition.
> > > >
> > > > We also tested a stricter variant where LoRA 3’s A→C training and evaluation pairs do not overlap in entities: even with additional chain-of-thought supervision to familiarize LoRA 3 with the question patterns, at test time it must route to LoRA 1 or 2 to recover the fact. In this setting, performance remains very poor and is roughly the same as using only the combination of LoRA 1 + 2, indicating that the extra parameters in LoRA 3 do not help and that simple reuse still fails to produce reliable A→C generalization. Together, these results suggest that our main negative findings are not an artifact of increasing LoRA parameter counts, but stem from the inherent difficulty of composing single-hop adapters into a new two-hop mapping. We will add this result to the paper.
> > > >
> > > > > In A.1, Eq (5), why is the current token not allowed to attend to itself? Casual language models often allow the tokens to attend to all past and itself. It also appears like the analysis relies on this fact.
> > > >
> > > > Indeed, we make this assumption to simplify the expressions in the proofs. The setup where the current token attends to itself leads to somewhat more complicated expressions, but without substantive change to the proofs. We have added a paragraph ``Role of Strict Causal Masking'' after the proof of Theorem 2 in Appendix A.1 to make this explicit.
> > > >
> > > >
> > > > > Would implications of LoRA on W in section 3 mean anything for LoRA on attention/MLP blocks in transformers? Transformers may rely on a completely different mechanism than LoRA on singled out linear layers. For example, the LoRA parameters of the two feedforward layers in MLP blocks of transformers could be coupled and work in tandem.
> > > >
> > > >
> > > > We agree that the theoretical analysis does not in principle preclude more complex interactions in multilayer models.
> > > > The motivation for our theoretical analysis comes from (i) the most relevant theoretical model of factual recall in transformers, by Nichani et al (2025, [1], cited in the paper), (ii) relevant mechanistic work on editing factual knowledge via a low-dimensional update to an MLP, especially ROME (Meng et al 2022, [2], cited in the paper).
> > > >
> > > > We agree that a full theoretical understanding in the multi-layer setup would be desirable. However, as we now explain in Appendix A.1.2, such understanding is unlikely to be attainable with existing methods. In fact, our analysis relies on similar simplifications as [1]. We also now acknowledge in the abstract that the theoretical analysis applies to a simplified setup.
> > > >
> > > > That said, we would like to emphasize that our experiments in Section 4 serve the purpose of verifying that conclusions suggested by our theoretical analysis do hold in the setting of real-world LMs with many layers.

---

> > > > > ### Author Response · Authors · 2025-12-13
> > > > > **Answers to additional questions**
> > > > >
> > > > > > Why is the proposed theoretical model a good proxy for a real transformer? It appears to be an extremely simplified version of a transformer. While analyzing an actual transformer might be practically intractable, I think there should be convincing explanation for why the results from these simplified models transfer to real models.
> > > > >
> > > > > Indeed, we acknowledge that the theoretical analysis applies to a highly simplified model. Our simplification is closely grounded in Nichani et al 2025 [1]. We have added Appendix A.1.2 to explain and motivate this. Taken together, our simplified one-layer model appears to be about as general as possible while making theoretical understanding of LoRA updates on our prompts feasible given available techniques.
> > > > >
> > > > > Our theoretical analysis is further motivated by intepretability work that indicates that MLPs perform factual recall. Our analysis suggests that applying LoRA reuse to any single such MLP will not unlock composition. We do acknowledge that our model does not rule out that interactions between different layers could somehow enable composition; rigorously ruling this out is likely beyond the scope of current theoretical methods.
> > > > > Importantly, experiments in Section 4 show that the key conclusion suggested by the theoretical analysis (failure of composition of LoRA adapters) also holds in real-world LLMs, even though they have far more layers.
> > > > > We now discuss this at the beginning of Section 3.
> > > > >
> > > > >
> > > > > > “We follow this assumption, and additionally take U, V to remain untrained, as we do not assume noise tokens in the context.” What does this mean?
> > > > >
> > > > > As mentioned above, we have (i) removed the reference to Nichani et al's "noise tokens", (ii) explicitly say that we also take K, Q to remain untrained.
> > > > >
> > > > > ---
> > > > > [1] Eshaan Nichani, Jason D. Lee, and Alberto Bietti. Understanding factual recall in transformers via associative memories. In The Thirteenth International Conference on Learning Representations, 2025.
> > > > >
> > > > > [2] Kevin Meng, David Bau, Alex Andonian, and Yonatan Belinkov. Locating and editing factual associations in GPT. Advances in neural information processing systems, 35:17359–17372, 2022

---

### Decision · Action_Editor_7qAj · 2026-01-03

**Recommendation:** Accept with minor revision

**Additional Comments:**

The authors are encouraged to revise the submission to improve the readability as per the reviewers' comments. In addition, please proofread the manuscript to eliminate various typos. For example,

1) Page 2, "...can be effective in data-constrained settings and **stresses** that..."
2) Page 3, "...and subsequently merges adapters by concatenating their weights that explicitly **leverages** held-out..."

**Audience:**

Yes

**Audience Explanation:**

LoRA is a widely used parameter-efficient finetuning method. Therefore, a comprehensive study of the limitations of its reuse to enable compositional generalization should be of interest to a large segment of TMLR’s audience.

**Claims And Evidence:**

Yes

**Claims Explanation:**

The paper studies compositional generalization via LoRA reuse, where the goal is to combine multiple LoRAs trained on distinct tasks to solve novel (compositional) tasks without requiring access to training data for those novel tasks. By analyzing a simplified setup inspired by Transformer architectures, the authors show that LoRA reuse cannot lead to compositional generalization.

The authors then conduct a systematic empirical study involving real-world LLMs of varying sizes and synthetic multi-hop reasoning and math tasks. Their study convincingly demonstrates the failure of compositional generalization via LoRA reuse unless the "bridges" (templates for compositional tasks) are already present in the training sets of the individual LoRAs.

During the review process, reviewers raised concerns about the synthetic evaluation tasks. In their rebuttal, the authors provided additional experiments on cross-lingual code generation and history/geography QA. These experiments further strengthen the contributions of the paper and provide robust empirical evidence for the submission's main takeaways.